# Low-Budget Simulation-Based Inference with Bayesian Neural Networks

## Abstract

Simulation-based inference methods have been shown to be inaccurate in the data-poor regime, when training simulations are limited or expensive. Under these circumstances, the inference network is particularly prone to overfitting, and using it without accounting for the computational uncertainty arising from the lack of identifiability of the network weights can lead to unreliable results. To address this issue, we propose using Bayesian neural networks in low-budget simulation-based inference, thereby explicitly accounting for the computational uncertainty of the posterior approximation. We design a family of Bayesian neural network priors that are tailored for inference and show that they lead to **better calibrated posteriors than standard methods** on tested benchmarks, even when as few as $O(10)$ simulations are available. This opens up the possibility of performing reliable simulation-based inference using very expensive simulators, as we demonstrate on a problem from the field of cosmology where single simulations are computationally expensive. We show that Bayesian neural networks produce informative and well-calibrated posterior estimates with only a few hundred simulations.

## 1 Introduction

Simulation-based inference aims at identifying the parameters of a stochastic simulator that best explain an observation. In its Bayesian formulation, simulation-based inference approximates the posterior distribution of the model parameters given an observation. This approximation usually takes the form of a neural network trained on synthetic data generated from the simulator. In the context of scientific discovery, Hermans et al. (2022) stressed the need for posterior approximations that are conservative – not overconfident – in order to make reliable downstream claims. They also showed that common simulation-based inference algorithms can produce overconfident approximations that may lead to erroneous conclusions.

In the data-poor regime (Villaescusa-Navarro et al., 2020; Zhang & Mikelsons, 2023; Zeng et al., 2023), where the simulator is expensive to run and only a small number of simulations are available, training a neural network to approximate the posterior can easily lead to overfitting. With small amounts of training data, the neural network weights are only loosely constrained, leading to high computational uncertainty (Wenger et al., 2022). That is, many neural networks can fit the training data equally well, yet they may have very different predictions on test data. For this reason, the posterior approximation is uncertain and, in the absence of a proper quantification of this uncertainty, potentially overconfident. Fortunately, computational uncertainty in a neural network can be quantified using Bayesian neural networks (BNNs) (Gal et al., 2016), which account for the uncertainty in the neural network weights. Therefore, in the context of simulation-based inference, BNNs can provide a principled way to quantify the computational uncertainty of the posterior approximation. Lueckmann et al. (2017) also make use of BNNs to iteratively refine a model on new data without having to retrain on old data.

Hermans et al. (2022) showed empirically that using ensembles of neural networks, a crude approximation of BNNs (Lakshminarayanan et al., 2017), does improve the calibration of the posterior approximation. A few studies have also used BNNs as density estimators in simulation-based inference (Cobb et al., 2019; Walmsley et al., 2020; Lemos et al., 2023). However, these studies have remained empirical and limited in their evaluation. This lack of theoretical grounding motivates the need for a more principled understanding of BNNs for simulation-based inference. In particular, the

choice of prior on the neural network weights happens to be crucial in this context, as it can strongly influence the resulting posterior approximation. Yet, arbitrary priors that convey little or undesired information about the posterior density have been used so far.

Our contributions are twofold. We first demonstrate both theoretically and empirically that, due to the prior on weights used, earlier attempts at simulation-based inference with Bayesian neural networks (Lueckmann et al., 2017; Cobb et al., 2019; Walmsley et al., 2020; Lemos et al., 2023) are inadequate for reliable inference. This motivates our second contribution, which is the design of an adequate prior on neural network's weights in the context of simulation-based inference. We show empirically that Bayesian neural networks equipped with this prior produce calibrated posteriors in the low-data regime. To our knowledge, this is the first method that provides reliable inference in that regime. The code is available at `https://github.com/anonymous`.

## 2 BACKGROUND

**Simulation-based inference**  We consider a stochastic simulator that takes parameters $\boldsymbol{\theta}$ as input and produces synthetic observations $\boldsymbol{x}$ as ouput. The simulator implicitly defines the likelihood $p(\boldsymbol{x}|\boldsymbol{\theta})$ in the form of a forward stochastic generative model but does not allow for direct evaluation of its density due to the intractability of the marginalization over its latent variables. In this setup, Bayesian simulation-based inference aims at approximating the posterior distribution $p(\boldsymbol{\theta}|\boldsymbol{x})$ using the simulator. Among possible approaches, *neural* simulation-based inference methods train a neural network to approximate key quantities from simulated data, such as the posterior, the likelihood, the likelihood-to-evidence ratio, or a score function (Cranmer et al., 2020).

Recently, concerns have been raised regarding the calibration of the approximate posteriors obtained with neural simulation-based inference. Hermans et al. (2022) showed that, unless special care is taken, common inference algorithms can produce overconfident posterior approximations. They quantify the calibration using the expected coverage

$$\text{EC}(\hat{p}, \alpha) = \mathbb{E}_{p(\boldsymbol{\theta}, \boldsymbol{x})}[\mathbb{1}(\boldsymbol{\theta} \in \boldsymbol{\Theta}_{\hat{p}}(\alpha))] \tag{1}$$

where $\boldsymbol{\Theta}_{\hat{p}}(\alpha)$ denotes the highest posterior credible region at level $\alpha$ computed using the posterior approximate $\hat{p}(\boldsymbol{\theta}|\boldsymbol{x})$. The expected coverage is equal to $\alpha$ when the posterior approximate is calibrated, lower than $\alpha$ when it is overconfident and higher than $\alpha$ when it is underconfident or conservative.

The calibration of posterior approximations has been improved in recent years in various ways. Delaunoy et al. (2022; 2023) regularize the posterior approximations to be balanced, which biases them towards conservative approximations. Similarly, Falkiewicz et al. (2024) regularize directly the posterior approximation by penalizing miscalibration or overconfidence. Masserano et al. (2023) use Neyman constructions to produce confidence regions with approximate Frequentist coverage. Patel et al. (2023) combine simulation-based inference and conformal predictions. Schmitt et al. (2023) enforce the self-consistency of likelihood and posterior approximations to improve the quality of approximate inference in low-data regimes.

**Bayesian deep learning**  Bayesian deep learning aims to account for both the aleatoric and epistemic uncertainty in neural networks. The aleatoric uncertainty refers to the intrinsic randomness of the variable being modeled, typically taken into account by switching from a point predictor to a density estimator. The epistemic uncertainty, on the other hand, refers to the uncertainty associated with the neural network itself and is typically high in small-data regimes. Failing to account for this uncertainty can lead to high miscalibration as many neural networks can fit the training data equally well, yet they may have very different predictions on test data.

Bayesian deep learning accounts for epistemic uncertainty by treating the neural network weights as random variables and considering the full posterior over possible neural networks instead of only the most probable neural network (Papamarkou et al., 2024). Formally, let us consider a supervised learning setting in all generality, where $\boldsymbol{x}$ denotes inputs, $\boldsymbol{y}$ outputs, $\boldsymbol{D}$ a dataset of $N$ pairs $(\boldsymbol{x}, \boldsymbol{y})$, and $\mathbf{w}$ the weights of the neural network. The likelihood of a given set of weights is

$$p(\boldsymbol{D}|\mathbf{w}) \propto \prod_{i=1}^{N} p(\boldsymbol{y}_i|\boldsymbol{x}_i, \mathbf{w}), \tag{2}$$

where $p(\boldsymbol{y}_i|\boldsymbol{x}_i, \mathbf{w})$ is the output of the neural network with weights $\mathbf{w}$ and inputs $\boldsymbol{x}_i$. The resulting posterior over the weights is

$$p(\mathbf{w}|\boldsymbol{D}) = \frac{p(\boldsymbol{D}|\mathbf{w})p(\mathbf{w})}{p(\boldsymbol{D})}, \tag{3}$$

where $p(\mathbf{w})$ is the prior. Once estimated, the posterior over the neural network's weights can be used for predictions through the Bayesian model average

$$p(\boldsymbol{y}|\boldsymbol{x}, \boldsymbol{D}) = \int p(\boldsymbol{y}|\boldsymbol{x}, \mathbf{w})p(\mathbf{w}|\boldsymbol{D})d\mathbf{w} \simeq \frac{1}{M}\sum_{i=1}^{M} p(\boldsymbol{y}|\boldsymbol{x}, \mathbf{w}_i), \mathbf{w}_i \sim p(\mathbf{w}|\boldsymbol{D}). \tag{4}$$

In practice, the Bayesian model average can be approximated by Monte Carlo sampling, with $M$ samples from the posterior over the weights. The quality of the approximation depends on the number of samples $M$, which should be chosen high enough to obtain a good enough approximation but small enough to keep reasonable the computational costs of predictions.

Estimating the posterior over the neural network weights is a challenging problem due to the high dimensionality of the weights. Variational inference (Blundell et al., 2015) optimizes a variational family to match the true posterior, which is typically fast but requires specifying a variational family that may restrict the functions that can be modeled. Markov chain Monte Carlo methods (Welling & Teh, 2011; Chen et al., 2014), on the other hand, are less restrictive in the functions that can be modeled but require careful tuning of the hyper-parameters and are more computationally demanding. The Bayesian posterior can also be approximated by an ensemble of neural networks (Lakshminarayanan et al., 2017; Pearce et al., 2020; He et al., 2020). Laplace methods leverage geometric information about the loss to construct an approximation of the posterior around the maximum a posteriori (MacKay, 1992). Similarly, Maddox et al. (2019) use the training trajectory of stochastic gradient descent to build an approximation of the posterior. **In this section, we propose a novel SBI algorithm based on Bayesian neural networks with tuned prior on weights. The algorithm first optimizes a prior on weight to have desirable conservativeness properties. This tuned prior is then used to compute an approximate posterior on weights that is itself used to make predictions through Bayesian model averaging.**

## 3 BAYESIAN NEURAL NETWORKS FOR SIMULATION-BASED INFERENCE

In the context of simulation-based inference, treating the weights of the inference network as random variables enables the quantification of the computational uncertainty of the posterior approximation. Considering neural networks taking observations $\boldsymbol{x}$ as input and producing parameters $\theta$ as output, the posterior approximation $\hat{p}(\boldsymbol{\theta}|\boldsymbol{x})$ can be modeled as the Bayesian model average

$$\hat{p}(\boldsymbol{\theta}|\boldsymbol{x}) = \int p(\boldsymbol{\theta}|\boldsymbol{x}, \mathbf{w})p(\mathbf{w}|\boldsymbol{D})d\mathbf{w}, \tag{5}$$

where $p(\boldsymbol{\theta}|\boldsymbol{x}, \mathbf{w})$ is the posterior approximation parameterized by the weights $\mathbf{w}$ and evaluated at $(\boldsymbol{\theta}, \boldsymbol{x})$, and $p(\mathbf{w}|\boldsymbol{D})$ is the posterior over the weights given the training set $\boldsymbol{D}$.

Remaining is the choice of prior $p(\mathbf{w})$. While progress has been made in designing better priors in Bayesian deep learning (Fortuin, 2022), we argue that none of those are suitable in the context of simulation-based inference. To illustrate our point, let us consider the case of a normal prior $p(\mathbf{w}) = \mathcal{N}(\mathbf{0}, \sigma^2 \boldsymbol{I})$ on the weights, in which case

$$\hat{p}_{\text{normal prior}}(\boldsymbol{\theta}|\boldsymbol{x}) = \int p(\boldsymbol{\theta}|\boldsymbol{x}, \mathbf{w}) \, \mathcal{N}(\mathbf{w}|\boldsymbol{\mu} = \mathbf{0}, \boldsymbol{\Sigma} = \sigma^2 \boldsymbol{I})d\mathbf{w}. \tag{6}$$

As mentioned in Section 2, a desirable property for a posterior approximation is to be calibrated. Therefore we want $\text{EC}(\hat{p}_{\text{normal prior}}, \alpha) = \alpha, \forall \alpha$. Although it might be possible for this property to be satisfied in particular settings, this is obviously not the case for all values of $\sigma$ and all neural network architectures. Therefore, and as illustrated in Figure 1, the Bayesian model average is not even calibrated a priori when using a normal prior on the weights. **This means that the Bayesian model average computed using the prior normal on weights $p(w)$ is not calibrated.** As the Bayesian model average is not calibrated a priori, it cannot be expected that updating the posterior over weights $p(\mathbf{w}|\boldsymbol{D})$ with a small amount of data would lead to a calibrated a posteriori Bayesian model average.

### 3.1 FUNCTIONAL PRIORS FOR SIMULATION-BASED INFERENCE

Our first contribution is the design of a prior that induces an a priori-calibrated Bayesian model average. To achieve this, we work in the space of posterior functions instead of the space of weights. We consider the space of functions taking a pair $(\boldsymbol{\theta}, \boldsymbol{x})$ as input and producing a posterior density value $f(\boldsymbol{\theta}, \boldsymbol{x})$ as output. Each function $f$ is defined by the joint outputs it associates with any arbitrary set of inputs, such that a posterior over functions can be viewed as a distribution over joint outputs for arbitrary inputs. Formally, let us consider $M$ arbitrary pairs $(\boldsymbol{\theta}, \boldsymbol{x})$ represented by the matrices $\boldsymbol{\Theta} = [\boldsymbol{\theta}_1, ..., \boldsymbol{\theta}_M]$ and $\boldsymbol{X} = [\boldsymbol{x}_1, ..., \boldsymbol{x}_M]$ and let $\boldsymbol{f} = [f_1, ..., f_M]$ be the joint outputs associated with those inputs. The distribution $p(\boldsymbol{f}|\boldsymbol{\Theta}, \boldsymbol{X})$ then represents a distribution over posteriors $\boldsymbol{f} = [\tilde{p}(\boldsymbol{\theta}_1|\boldsymbol{x}_1), ..., \tilde{p}(\boldsymbol{\theta}_M|\boldsymbol{x}_M)]$. The functional posterior distribution over posteriors for parameters $\boldsymbol{\Theta}$ and observations $\boldsymbol{X}$ given a training dataset $\boldsymbol{D}$ is then $p(\boldsymbol{f}|\boldsymbol{\Theta}, \boldsymbol{X}, \boldsymbol{D})$ and the Bayesian model average is obtained through marginalization, that is

$$p(\boldsymbol{\theta}_i|\boldsymbol{x}_i, \boldsymbol{D}) = \int f_i \, p(\boldsymbol{f}|\boldsymbol{\Theta}, \boldsymbol{X}, \boldsymbol{D})d\boldsymbol{f}, \quad \forall i. \tag{7}$$

Computing the posterior over functions requires the specification of a prior over functions. We first observe that the prior over the simulator's parameters is a calibrated approximation of the posterior. That is, for the prior function $p_{\text{prior}} : (\boldsymbol{\theta}, \boldsymbol{x}) \to p(\boldsymbol{\theta})$, we have that $\text{EC}(p_{\text{prior}}, \alpha) = \alpha, \forall \alpha$ (Delaunoy et al., 2023). It naturally follows that the a priori Bayesian model average with a Dirac delta prior around the prior on the simulator's parameters is calibrated

$$\hat{p}(\boldsymbol{\theta}_i|\boldsymbol{x}_i) = \int f_i \, \delta([f_j = p_{\text{prior}}(\boldsymbol{\theta}_j, \boldsymbol{x}_j)]) \, d\boldsymbol{f}, \forall i$$
$$= \int f_i \, \delta(f_i = p(\boldsymbol{\theta}_i)) \, df_i, \forall i \Rightarrow \text{EC}(\hat{p}, \alpha) = \alpha, \forall \alpha. \tag{8}$$

However, this prior has limited support, and the Bayesian model average will not converge to the posterior $p(\boldsymbol{\theta}|\boldsymbol{x})$ as the dataset size increases. We extend this Dirac prior to include more functions in its support while retaining the calibration property, which we propose defining as a Gaussian process centered at $p_{\text{prior}}$.

A Gaussian process (GP) defines a joint multivariate normal distribution over all the outputs $\boldsymbol{f}$ given the inputs $(\boldsymbol{\Theta}, \boldsymbol{X})$. It is parametrized by a mean function $\mu$ that defines the mean value for the outputs given the inputs and a kernel function $K$ that models the covariance between the outputs. If we have access to no data, the mean and the kernel jointly define a prior over functions as they define a joint prior over outputs for an arbitrary set of inputs. In order for this prior over functions to be centered around the prior $p_{\text{prior}}$, we define the mean function as $\mu(\boldsymbol{\theta}, \boldsymbol{x}) = p(\boldsymbol{\theta})$. The kernel $K$, on the other hand, defines the spread around the mean function and the correlation between the outputs $\boldsymbol{f}$. Its specification is application-dependent and constitutes a hyper-parameter of our method that can be exploited to incorporate domain knowledge on the structure of the posterior. For example, periodic kernels could be used if periodicity is observed. Kernel's hyperparameters can also be chosen such as to incorporate what would be a reasonable deviation of the approximated posterior from the prior. We denote the Gaussian process prior over function outputs as $p_{\text{GP}}(\boldsymbol{f}|\mu(\boldsymbol{\Theta}, \boldsymbol{X}), K(\boldsymbol{\Theta}, \boldsymbol{X}))$. Proposition 1 shows that a functional prior defined in this way leads to a calibrated Bayesian model average.

**Proposition 1.** *The Bayesian model average of a Gaussian process centered around the prior on the simulator's parameters is calibrated. Formally, let $p_{GP}$ be the density probability function defined by a Gaussian process, $\mu$ its mean function, and $K$ the kernel. Let us consider $M$ arbitrary pairs $(\boldsymbol{\theta}, \boldsymbol{x})$ represented by the matrices $\boldsymbol{\Theta} = [\boldsymbol{\theta}_1, ..., \boldsymbol{\theta}_M]$ and $\boldsymbol{X} = [\boldsymbol{x}_1, ..., \boldsymbol{x}_M]$ and represent by the vector $\boldsymbol{f} = [f_1, ..., f_M]$ the joint outputs associated with those inputs. The Bayesian model average on the $i^{th}$ pair is expressed*

$$\hat{p}(\boldsymbol{\theta}_i|\boldsymbol{x}_i) = \int f_i \, p_{GP}(\boldsymbol{f}|\mu(\boldsymbol{\Theta}, \boldsymbol{X}), K(\boldsymbol{\Theta}, \boldsymbol{X})) \, d\boldsymbol{f}$$

*If $\mu(\boldsymbol{\theta}, \boldsymbol{x}) = p(\boldsymbol{\theta}), \forall \boldsymbol{\theta}, \boldsymbol{x}$, then,*
$$\text{EC}(\hat{p}, \alpha) = \alpha, \forall \alpha,$$

*for all kernel $K$.*

*Proof.* As $p_{\text{GP}}$ is, by definition of a Gaussian process, a multivariate normal, the expectations of the marginals are equal to the mean parameters

$$\hat{p}(\boldsymbol{\theta}_i|\boldsymbol{x}_i) = \mu(\boldsymbol{\theta}_i, \boldsymbol{x}_i) = p(\boldsymbol{\theta}_i).$$

The joint evaluation of the Bayesian model average of the Gaussian process is hence equivalent to the joint evaluation of the prior for any matrices $\boldsymbol{\Theta}$ and $\boldsymbol{X}$. We can therefore conclude that $\hat{p}$ is equivalent to $p_{\text{prior}} : (\boldsymbol{\theta}, \boldsymbol{x}) \to p(\boldsymbol{\theta})$. Since $\text{EC}(p_{\text{prior}}, \alpha) = \alpha, \forall \alpha$ (Delaunoy et al., 2023), then, $\text{EC}(\hat{p}, \alpha) = \alpha, \forall \alpha$. $\qquad\square$

## 3.2 FROM FUNCTIONAL TO PARAMETRIC PRIORS

In this section, we now discuss how existing work from Bayesian deep learning in function space can be used to perform simulation-based inference with the functional GP prior over posterior density functions proposed in Section 3.1. We follow Flam-Shepherd et al. (2017) and Sun et al. (2018) for mapping the functional prior to a distribution over neural network weights, but we note that other methods for functional Bayesian deep learning, such as those presented by Tran et al. (2022); Rudner et al. (2022); Kozyrskiy et al. (2023); Ma & Hernández-Lobato (2021) could also be used in our setting. Further discussion can be found in Appendix A.

Let us first observe that a neural network architecture and a prior on weights jointly define a prior over functions. We parameterize the prior on weights by $\phi$ and denote this probability density over function outputs by

$$
\begin{aligned}
p_{\text{BNN}}(\boldsymbol{f} \mid \phi, \boldsymbol{\Theta}, \boldsymbol{X}) &= \int p(\boldsymbol{f} \mid \mathbf{w}, \boldsymbol{\Theta}, \boldsymbol{X}) \, p(\mathbf{w}|\phi) \, d\mathbf{w} \\
&= \int \delta([f_i = p(\boldsymbol{\theta}_i|\boldsymbol{x}_i, \mathbf{w})]) \, p(\mathbf{w}|\phi) \, d\mathbf{w}.
\end{aligned}
\tag{9}
$$

To obtain a prior on weights that matches the target GP prior, we optimize $\phi$ such that $p_{\text{BNN}}(\boldsymbol{f} \mid \phi, \boldsymbol{\Theta}, \boldsymbol{X})$ matches $p_{\text{GP}}(\boldsymbol{f}|\mu(\boldsymbol{\Theta}, \boldsymbol{X}), K(\boldsymbol{\Theta}, \boldsymbol{X}))$. Following Flam-Shepherd et al. (2017), given a measurement set $\mathcal{M} = \{\boldsymbol{\theta}_i, \boldsymbol{x}_i\}_{i=1}^{M}$ at which we want the distributions to match, the KL divergence between the two priors can be expressed as

$$
\begin{aligned}
&\text{KL}\left[p_{\text{BNN}}(\boldsymbol{f} \mid \phi, \mathcal{M}) \, \| \, p_{\text{GP}}(\boldsymbol{f} \mid \mu(\mathcal{M}), K(\mathcal{M}))\right] \\
&= \int p_{\text{BNN}}(\boldsymbol{f} \mid \phi, \mathcal{M}) \log \frac{p_{\text{BNN}}(\boldsymbol{f} \mid \phi, \mathcal{M})}{p_{\text{GP}}(\boldsymbol{f} \mid \mu(\mathcal{M}), K(\mathcal{M}))} d\boldsymbol{y} \\
&= -\mathbb{H}\left[p_{\text{BNN}}(\boldsymbol{f} \mid \phi, \mathcal{M})\right] - \mathbb{E}_{p_{\text{BNN}}(\boldsymbol{f} \mid \phi, \mathcal{M})}\left[\log p_{\text{GP}}(\boldsymbol{f} \mid \mu(\mathcal{M}), K(\mathcal{M}))\right],
\end{aligned}
\tag{10}
$$

where the second term $\mathbb{E}_{p_{\text{BNN}}(\boldsymbol{f} \mid \phi, \mathcal{M})}\left[\log p_{\text{GP}}(\boldsymbol{f} \mid \mu(\mathcal{M}), K(\mathcal{M}))\right]$ can be estimated using Monte-Carlo. The first term $\mathbb{H}\left[p_{\text{BNN}}(\boldsymbol{f} \mid \phi, \mathcal{M})\right]$, however, is harder to estimate as it requires computing $\log p_{\text{BNN}}(\boldsymbol{f} \mid \phi, \mathcal{M})$, which involves the integration of the output over all possible weights combinations. To bypass this issue, Sun et al. (2018) propose to use Spectral Stein Gradient Estimation (SSGE) (Shi et al., 2018) to approximate the gradient of the entropy as

$$\nabla\mathbb{H}\left[p_{\text{BNN}}(\boldsymbol{f} \mid \phi, \mathcal{M})\right] \simeq \text{SSGE}\left(\boldsymbol{f}_1, ..., \boldsymbol{f}_N \sim p_{\text{BNN}}(\boldsymbol{f} \mid \phi, \mathcal{M})\right). \tag{11}$$

We note that the measurement set $\mathcal{M}$ can be chosen arbitrarily but should cover most of the support of the joint distribution $p(\boldsymbol{\theta}, \boldsymbol{x})$. If data from this joint distribution are available, those can be leveraged to build the measurement set. To showcase the ability to create a prior with limited data, in this work, we derive boundaries of the support of each marginal distribution and draw parameters and observations independently and uniformly over this support. If the support is known a priori, this procedure can be performed without (expensive) simulations. We draw a new measurement set at each iteration of the optimization procedure. If a fixed measurement set is available, a subsample of this measurement set should be drawn at each iteration.

As an illustrative example, we chose independent normal distributions as a variational family $p(\mathbf{w}|\phi)$ over the weights and minimize (10) w.r.t. $\mathbf{w}$. In Figure 1, we show the coverage of the resulting a priori Bayesian model average using the tuned prior, $p(\mathbf{w} \mid \phi)$, and normal priors for increasing standard deviations $\sigma$, for the SLCP benchmark. We observe that while none of the normal priors are calibrated, the trained prior achieves near-perfect calibration. This prior hence guides the

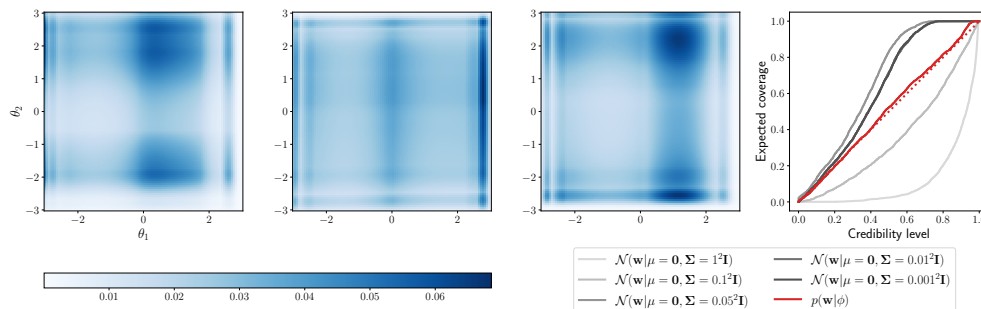

Figure 1: Visualization of the prior **over neural network's weights** tuned to match the GP prior on the SLCP benchmark. Left: examples of posterior functions **over simulator's parameters** sampled from the tuned prior over neural network's weights. Right: expected coverage of the prior Bayesian model average with the tuned prior and normal priors for varying standard deviations.

obtained posterior approximation towards more calibrated solutions, even in low simulation-budget settings.

The attentive reader might have noticed that $p_{\text{BNN}}(\boldsymbol{f} \mid \boldsymbol{\phi}, \boldsymbol{\Theta}, \boldsymbol{X})$ and $p_{\text{GP}}(\boldsymbol{f}|\mu(\boldsymbol{\Theta}, \boldsymbol{X}), K(\boldsymbol{\Theta}, \boldsymbol{X}))$ do not share the same support, as the former distribution is limited to functions that represent valid densities by construction, while the latter includes arbitrarily shaped functions. This is not an issue here as the support of the first distribution is included in the support of the second distribution, and functions outside the support of the first distribution are ignored in the computation of the divergence.

## 4 EXPERIMENTS

In this section, we empirically demonstrate the benefits of replacing a regular neural network with a BNN equipped with the proposed prior for simulation-based inference. We consider both Neural Posterior Estimation (NPE) with neural spline flows (Durkan et al., 2019) and Neural Ratio Estimation (NRE) (Hermans et al., 2020), along with their balanced versions (BNRE and BNPE) (Delaunoy et al., 2022; 2023) and ensembles (Lakshminarayanan et al., 2017; Hermans et al., 2022). BNNs-based methods are trained using mean-field variational inference (Blundell et al., 2015). As advocated by Wenzel et al. (2020), we also consider cold posteriors to achieve good predictive performance. More specifically, the variational objective function is modified to give less weight to the prior by introducing a temperature parameter $T$,

$$\mathbb{E}_{\mathbf{w}\sim p(\mathbf{w}|\boldsymbol{\tau})} \left[ \sum_i \log p(\boldsymbol{\theta}_i|\boldsymbol{x}_i, \mathbf{w}) \right] - T \, \text{KL}[p(\mathbf{w}|\boldsymbol{\tau})||p(\mathbf{w}|\boldsymbol{\phi})], \tag{12}$$

where $\boldsymbol{\tau}$ are the parameters of the posterior variational family and $T$ is a parameter called the temperature that weights the prior term. In the following, we call BNN-NPE, a Bayesian Neural Network posterior estimator trained without temperature ($T = 1$), and BNN-NPE ($T = 0.01$), an estimator trained with a temperature of $0.01$, assigning a lower weight to the prior.

A detailed description of the Gaussian process used can be found in Appendix A. For simplicity, in this analysis, we use an RBF kernel in the GP prior. If more information on the structure of the target posterior is available, more informed kernels may be used to leverage this prior knowledge. A description of the benchmarks can be found in Appendix B, and the hyperparameters are described in Appendix C.

Following Delaunoy et al. (2022), we evaluate the quality of the posterior approximations based on the expected nominal log posterior density and the expected coverage area under the curve (coverage AUC). The expected nominal log posterior density $\mathbb{E}_{\boldsymbol{\theta},\boldsymbol{x}\sim p(\boldsymbol{\theta},\boldsymbol{x})}[\log \hat{p}(\boldsymbol{\theta}|\boldsymbol{x})]$ quantifies the amount of density allocated to the nominal parameter that was used to generate the observation. The coverage AUC $\int_0^1 (\text{EC}(\hat{p}, \alpha) - \alpha) \, d\alpha$ quantifies the calibration of the expected posterior. A calibrated posterior

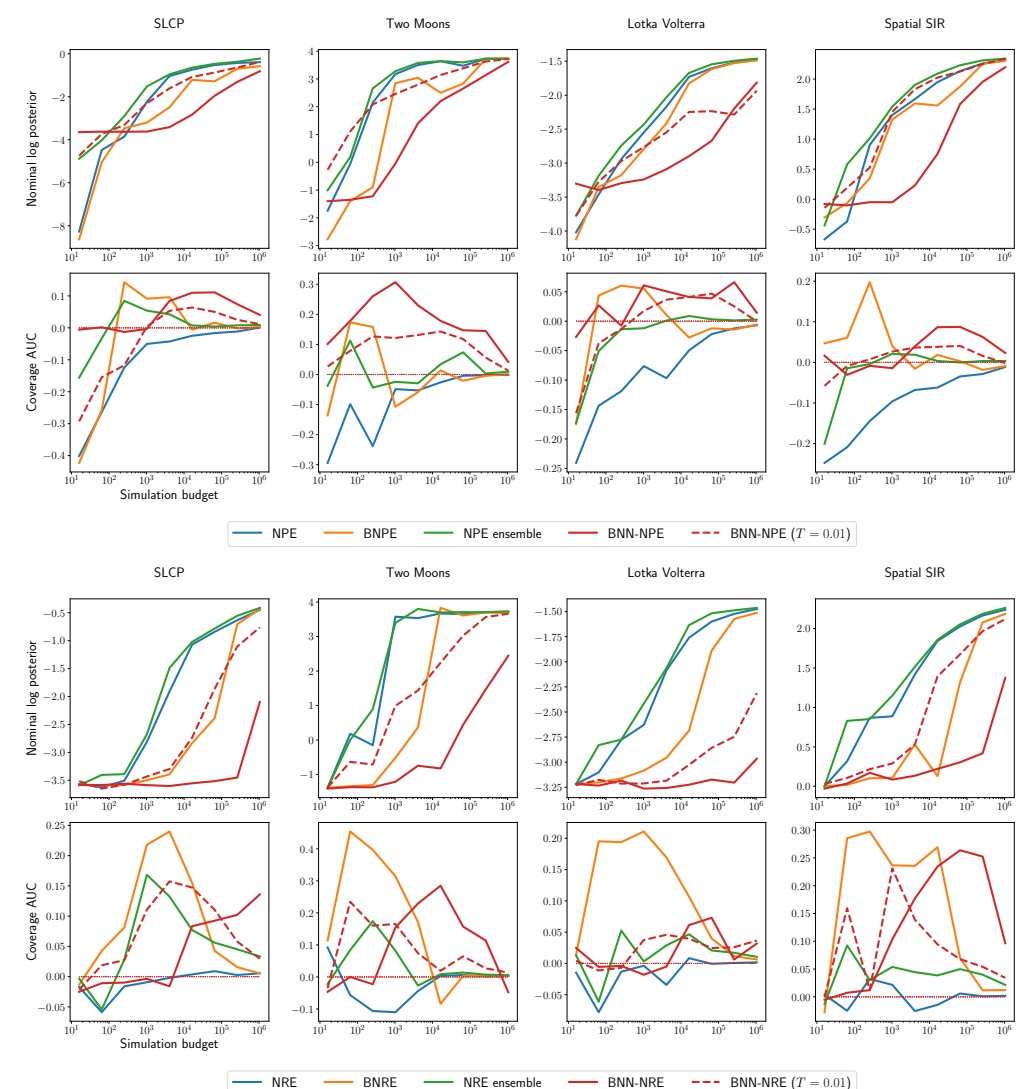

Figure 2: Comparison of simulation-based inference methods through the nominal log probability and coverage area under the curve. The higher the nominal log probability, the more performant the method is. A calibrated posterior approximation exhibits a coverage AUC of $0$. A positive coverage AUC indicates conservativeness, and a negative coverage AUC indicates overconfidence. 3 runs are performed, and the median is reported. The plot at the top shows the results for NPE simulation-based inference methods, and the one at the bottom shows NRE methods.

approximation exhibits a coverage AUC of $0$. A positive coverage AUC indicates conservativeness, and a negative coverage AUC indicates overconfidence.

**BNN-based simulation-based inference**    Figure 2 compares simulation-based inference methods with and without accounting for computational uncertainty. We observe that BNNs equipped with our prior and without temperature show positive, **or only slightly negative,** coverage AUC even for simulation budgets as low as $O(10)$. **Negative coverage AUC is still observed, and hence conservativeness is not strictly guaranteed. However, this constitutes a significant improvement over the other method in that regard.** The coverage curves are reported in Appendix D. We conclude that BNNs can hence be **more** reliably used **than the other benchmarked methods** when the simulator is expensive and few simulations are available. **We observe that increasing the reliability comes with the drawback of requiring more simulations than the other methods to reach similar nominal log posterior density values. Without temperature, a few orders of magnitude**

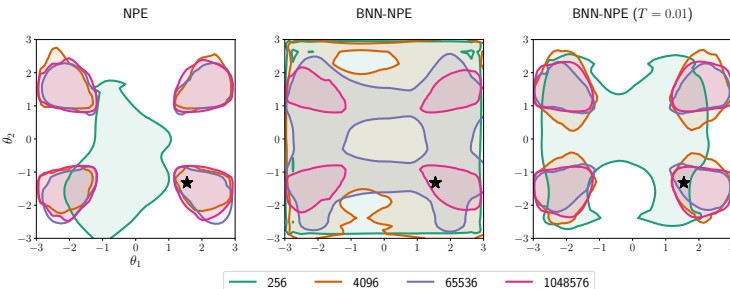

Figure 3: Examples of 95% highest posterior density regions obtained with various algorithms and simulation budgets on the SLCP benchmark for a single observation. The black star represents the ground truth used to generate the observation **and the legend indicates the simulation budget**.

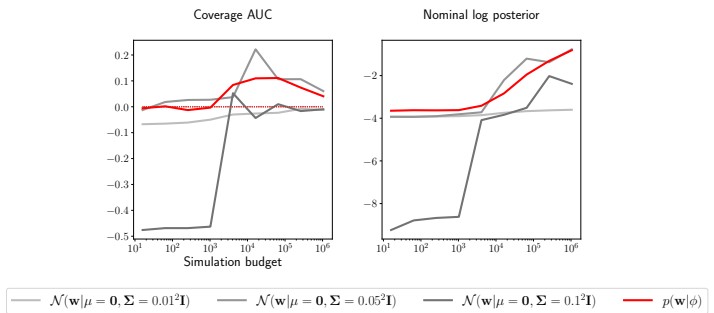

Figure 4: Comparison of posterior approximations obtained using a prior tuned to match the Gaussian process-based prior and using independent normal priors on weights with zero means and various standard deviations on the SLCP benchmark. 3 runs are performed, and the median is reported.

**more samples might be needed. However, in theory, as the amount of sample increases, the effect of the prior diminishes, and BNNs should reach the same nominal log posterior density as standard methods. By adding a temperature to the prior, its effect is diminished and better nominal log posterior density values are observed** From these observations, general guidelines to set the temperature include increasing $T$ if overconfidence is observed and decreasing it if low predictive performance is observed.

Examples of posterior approximations obtained with and without using a Bayesian neural network are shown in Figure 3. Wide posteriors are observed for low budgets for BNN-NPE, while NPE produces an overconfident approximation and excludes most of the relevant parts of the posterior. As the simulation budget increases, BNN-NPE converges slowly towards the same posterior as NPE. BNN-NPE ($T = 0.01$) converges faster than BNN-NPE but, for low simulation budgets, excludes parts of the region that should be accepted according to high budget posteriors. Yet, the posterior approximate is still less overconfident than NPE's. Finally, Figure 2 shows that BNN-NRE is more conservative than BNN-NPE. This comes at the cost of lower nominal log posterior density for a given simulation budget.

**Comparison of different priors on weights** We analyze the effect of the prior on the neural network's weights on the resulting posterior approximation. The posterior approximations obtained using our GP prior are compared to the ones obtained using independent normal priors on weights with zero means and increasing standard deviations. In Figure 4, we observe that when using a normal prior, careful tuning of the standard deviation is needed to achieve results close to the prior designed for simulation-based inference. The usage of an inappropriate prior can lead to bad calibration for low simulation budgets or can prevent learning if it is too restrictive.

**Uncertainty decomposition** We decompose the uncertainty quantified by the different methods. Following Depeweg et al. (2018), the uncertainty can be decomposed as

$$\mathbb{H}\left[\hat{p}(\boldsymbol{\theta}|\boldsymbol{x})\right] = \mathbb{E}_{q(\mathbf{w})}\left[\mathbb{H}\left[\hat{p}(\boldsymbol{\theta}|\boldsymbol{x}, \mathbf{w})\right]\right] + \mathbb{I}(\boldsymbol{\theta}, \mathbf{w}), \tag{13}$$

where $\mathbb{E}_{q(\mathbf{w})}\left[\mathbb{H}\left[\hat{p}(\boldsymbol{\theta}|\boldsymbol{x}, \mathbf{w})\right]\right]$ quantifies the aleatoric uncertainty, $\mathbb{I}(\boldsymbol{\theta}, \mathbf{w})$ quantifies the epistemic uncertainty, and the sum of those terms is the predictive uncertainty. Figure 5 shows the decomposition of the two sources of uncertainty, in expectation, on the SLCP benchmark. Other benchmarks can be found in Appendix D. We observe that BNN-NPE and NPE ensemble methods account for the epistemic uncertainty while other methods do not. BNPE artificially increases the aleatoric uncertainty to be better calibrated. The epistemic uncertainty of BNN-NPE is initially low because most of the models are slight variations of $p_{\boldsymbol{\Theta}}$. The epistemic uncertainty then increases as it starts to deviate from the prior and decreases as the training set size increases. BNN-NPE ($T = 0.01$) exhibits a higher epistemic uncertainty for low budgets as the effect of the prior is lowered.

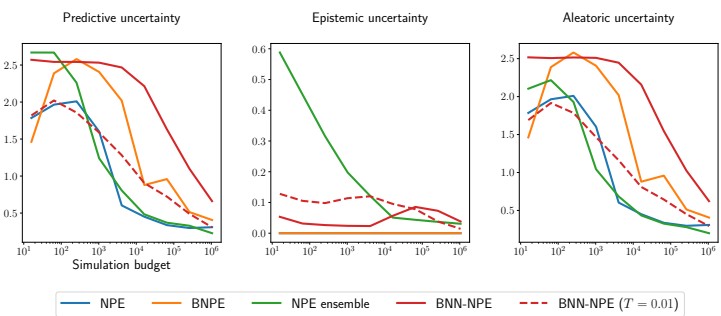

Figure 5: Quantification of the different forms of uncertainties captured by the different NPE-based methods on the SLCP benchmark. 3 runs are performed, and the median is reported.

**Inferring cosmological parameters from $N$-body simulations** To showcase the utility of Bayesian deep learning for simulation-based inference in a practical setting, we consider a challenging inference problem from the field of cosmology. We consider *Quijote $N$*-body simulations (Villaescusa-Navarro et al., 2020) tracing the spatial distribution of matter in the Universe for different underlying cosmological models. The resulting observations are particles with different masses, corresponding to dark matter clumps, which host galaxies. We consider the canonical task of inferring the matter density (denoted $\Omega_m$) and the root-mean-square matter fluctuation averaged over a sphere of radius $8h^{-1}$ Mpc (denoted $\sigma_8$) from an observed galaxy field. Robustly inferring the values of these parameters is one of the scientific goals of flagship cosmological surveys. These simulations are very computationally expensive to run, with over 35 million CPU hours required to generate 44100 simulations at a relatively low resolution. Generating samples at higher resolutions, or a significantly larger number of samples, is challenging due to computational constraints. These constraints necessitate methods that can be used to produce reliable scientific conclusions from a limited set of simulations – when few simulations are available, not only is the amount of training data low, but so is the amount of test data that is available to assess the calibration of the trained model.

In this experiment, we use 2000 simulations processed as described in Cuesta-Lazaro & Mishra-Sharma (2023). These simulations form a subset of the full simulation suite run with a uniform prior over the parameters of interest. 1800 simulations are used for training and 200 are kept for testing. We use the two-point correlation function evaluated at 24 distance bins as a summary statistic. The observable is, hence, a vector of 24 features. We observed that setting a temperature lower than 1 was needed to reach reasonable predictive performance with Bayesian neural networks in this setting. Figure 6 compares the posterior approximations obtained with a single neural network against those obtained with a BNN trained with a temperature of 0.01. We observe from the coverage plots that while a single neural network can lead to overconfident approximations in the data-poor regime, the BNN leads to conservative approximations. BNN-NPE also exhibits higher nominal log posterior probability. Additionally, we observe that it provides posterior approximations that are calibrated and have a high nominal log probability with only a few hundred samples.

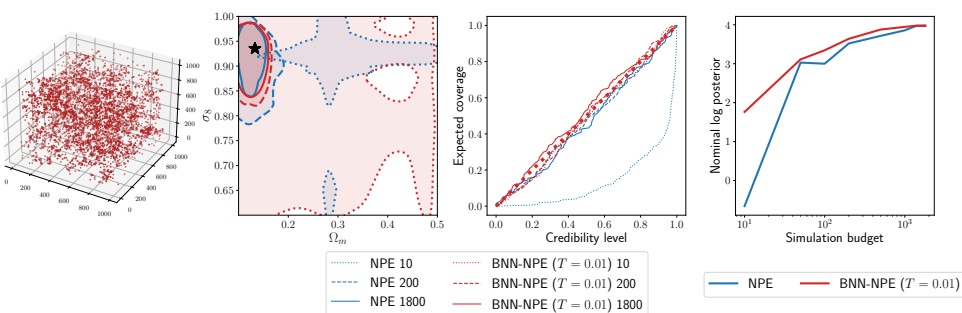

Figure 6: Comparison of the posterior approximations obtained with and without a Bayesian neural network on the cosmological application. First plot: An example observation: particles representing galaxies in a synthetic universe. Second plot: example of $95\%$ highest posterior density regions for increasing simulation budgets. The black star represents the ground truth used to generate the observation. Third plot: Expected coverage with and without using a Bayesian neural network for increasing simulation budgets. Fourth plot: The nominal log posterior.

## 5 CONCLUSION

In this work, we use Bayesian deep learning to account for the computational uncertainty associated with posterior approximations in simulation-based inference. We show that the prior on neural network's weights should be carefully chosen to obtain calibrated posterior approximations and develop a prior family with this objective in mind. The prior family is defined in function space as a Gaussian process and mapped to a prior on weights. Empirical results on benchmarks show that incorporating Bayesian neural networks in simulation-based inference methods consistently yields conservative posterior approximations, even with limited simulation budgets of $\mathcal{O}(10)$. As Bayesian deep learning continues to rapidly advance (Papamarkou et al., 2024), we anticipate that future developments will strengthen its applicability in simulation-based inference, ultimately enabling more efficient and reliable scientific applications in domains with computationally expensive simulators.

Using BNNs for simulation-based inference also comes with limitations. The first observed limitation is that the Bayesian neural network based methods might need **orders of magnitude** more simulated data in order to reach a predictive power similar to methods that do not use BNNs, such as NPE. While we showed that this limitation can be mitigated by tuning the temperature appropriately, this is something that might require trials and errors. A second limitation is the computational cost of predictions. When training a BNN using variational inference, the training cost remains on a similar scale as standard neural networks. At prediction time, however, the Bayesian model average described in Equation 4 must be approximated, and this requires a neural network evaluation for each Monte Carlo sample in the approximation. The computational cost of predictions then scales linearly with the number $M$ of Monte Carlo samples. **Finally, although our method significantly improves the reliability over standard methods for low simulation budgets, conservativeness is not strictly guaranteed. There are no theoretical guarantees and negative coverage AUC may still be observed.**

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

## A   PRIOR TUNING DETAILS

We tune the parameters $\phi$ of a variational distribution over neural network weights $p(\mathbf{w}|\phi)$. The variational distribution is chosen to be independent normal distributions, with parameters $\phi$ representing the means and standard deviations of each parameter of $\mathbf{w}$. This variational family defines a prior over function outputs

$$p_{\text{BNN}}(\boldsymbol{f} \mid \phi, \boldsymbol{\Theta}, \boldsymbol{X}) = \int p(\boldsymbol{f} \mid \mathbf{w}, \boldsymbol{\Theta}, \boldsymbol{X}) p(\mathbf{w}|\phi) d\mathbf{w}. \tag{14}$$

The parameters $\phi$ are optimized to obtain a prior on weights that matches the target Gaussian process functional prior $p_{\text{GP}}(\boldsymbol{f}|\mu(\boldsymbol{\Theta}, \boldsymbol{X}), K(\boldsymbol{\Theta}, \boldsymbol{X}))$. To achieve this, we repeatedly sample a measurement set $\mathcal{M} = \{\boldsymbol{\theta}_i, \boldsymbol{x}_i\}_{i=1}^{M}$ and $N$ function outputs from the BNN prior $\boldsymbol{f}_1, ..., \boldsymbol{f}_N \sim p_{\text{BNN}}(\boldsymbol{f} \mid \phi, \mathcal{M})$ and perform a step of gradient descend to minimize the divergence

$$\text{KL}\left[p_{\text{BNN}}(\boldsymbol{f} \mid \phi, \mathcal{M}) \,\|\, p_{\text{GP}}(\boldsymbol{f} \mid \mu(\mathcal{M}), K(\mathcal{M}))\right]. \tag{15}$$

The mean function $\mu$ of the Gaussian process is selected as:

$$\mu(\boldsymbol{\theta}, \boldsymbol{x}) = p(\boldsymbol{\theta}). \tag{16}$$

The kernel $K$ is a combination of two Radial Basis Function (RBF) kernels

$$K(\boldsymbol{\theta}_1, \boldsymbol{\theta}_2, \boldsymbol{x}_1, \boldsymbol{x}_2) = \sqrt{\text{RBF}(\boldsymbol{\theta}_1, \boldsymbol{\theta}_2)} * \sqrt{\text{RBF}(\boldsymbol{x}_1, \boldsymbol{x}_2)}. \tag{17}$$

such that the correlation between outputs is high only if $\boldsymbol{\theta}_1$ and $\boldsymbol{\theta}_2$ as well as $\boldsymbol{x}_1$ and $\boldsymbol{x}_2$ are close. The RBF kernel is defined as

$$\text{RBF}(\boldsymbol{x}_1, \boldsymbol{x}_2) = \sigma^2 \exp\left(-\frac{1}{N} \sum_i^N \frac{(x_{1,i} - x_{2,i})^2}{2l_i^2}\right), \tag{18}$$

where $\sigma$ is the standard deviation and $l_i$ is the lengthscale associated to the $i^{\text{th}}$ feature. The lengthscale is derived from the measurement set. To determine $l_i$, we query observations $\boldsymbol{x}$ from the measurement set and compute the 0.1 quantile of the squared distance between different observations for each feature. We then set $l_i$ such that $2l_i^2$ equals this quantile. All the benchmarks have a uniform prior over the simulator's parameters. The mean function is then equal to a constant $C$ for all input values. The standard deviation is chosen to be $C/2$. To ensure stability during the inference procedure, we enforce all standard deviations defined in $\phi$ to be at least 0.001 by setting any parameters below this threshold to this value.

Note that there are various methods that can be used to perform inference on the neural network's weights with our GP prior. Instead of minimizing the KL-divergence, the parameters $\phi$ can be

optimized using an adversarial training procedure by treating both priors as function generators and training a discriminator between the two (Tran et al., 2022). Another approach to performing inference using a functional prior is to directly use it during inference by modifying the inference algorithm to work in function space. Variational inference can be performed in the space of function (Sun et al., 2018; Rudner et al., 2022). The stochastic gradient Hamiltonian Monte Carlo algorithm (Chen et al., 2014) could also be modified to include a functional prior Kozyrskiy et al. (2023). Alternatively, a variational implicit process can be learned to express the posterior in function space (Ma & Hernández-Lobato, 2021).

## B  Benchmarks description

**SLCP**  The SLCP (Simple Likelihood Complex Posterior) benchmark (Papamakarios et al., 2019) is a fictive benchmark that takes 5 parameters as input and produces an 8-dimensional synthetic observable. The observation corresponds to the 2D coordinates of 4 points that are sampled from the same multivariate normal distribution. We consider the task of inferring the marginal over 2 of the 5 parameters.

**Two Moons**  The Two Moons simulator (Greenberg et al., 2019) models a fictive problem with 2 parameters. The observable x is composed of 2 scalars, which represent the 2D coordinates of a random point sampled from a crescent-shaped distribution shifted and rotated around the origin depending on the parameters' values. Those transformations involve the absolute value of the sum of the parameters leading to a second crescent in the posterior and, hence making it multi-modal.

**Lotka Volterra**  The Lotka-Volterra population model (Lotka, 1920; Volterra, 1926) describes a process of interactions between a predator and a prey species. The model is conditioned on 4 parameters that influence the reproduction and mortality rate of the predator and prey species. We infer the marginal posterior of the predator parameters from a time series of 2001 steps representing the evolution of both populations over time. The specific implementation is based on a Markov Jump Process, as in Papamakarios et al. (2019).

**SpatialSIR**  The Spatial SIR model (Hermans et al., 2022) involves a grid world of susceptible, infected, and recovered individuals. Based on initial conditions and the infection and recovery rate, the model describes the spatial evolution of an infection. The observable is a snapshot of the grid world after some fixed amount of time. The grid used is of size 50 by 50.

## C  Hyperparameters

All the NPE-based methods use a Neural Spline Flow (NSF) (Durkan et al., 2019) with 3 transforms of 6 layers, each containing 256 neurons. Meanwhile, all the NRE-based methods employ a classifier consisting of 6 layers of 256 neurons. For the spatialSIR and Lotka Volterra benchmarks, the observable is initially processed by an embedding network. Lotka Volterra's embedding network is a 10 layers 1D convolutional neural network that leads to an embedding of size 512. On the other hand, SpatialSIR's embedding network is an 8 layers 2D convolutional neural network resulting in an embedding of size 256. All the models are trained for 500 epochs which we observed to be enough to reach convergence. **In addition, all the training procedures make use of a validation set to control overfitting.**

Bayesian neural network-based methods use independent normal distributions as a variational family. During inference, 100 neural networks are sampled to approximate the Bayesian model average. Ensemble methods involve training 5 neural networks independently. The experiments were conducted on a private GPU cluster, and the estimated computational cost is around $25,000$ GPU hours.

## D  Additional experiments

In this section, we provide complementary results. Figures 7 and 8 display the coverage curves, demonstrating that a higher positive coverage AUC corresponds to coverage curves above the diagonal line. Figures 9 and 10 present the uncertainty decomposition of all methods on all the bench-

marks. Figures 11 and 12 illustrate how the performance of the different algorithms varies across different runs.

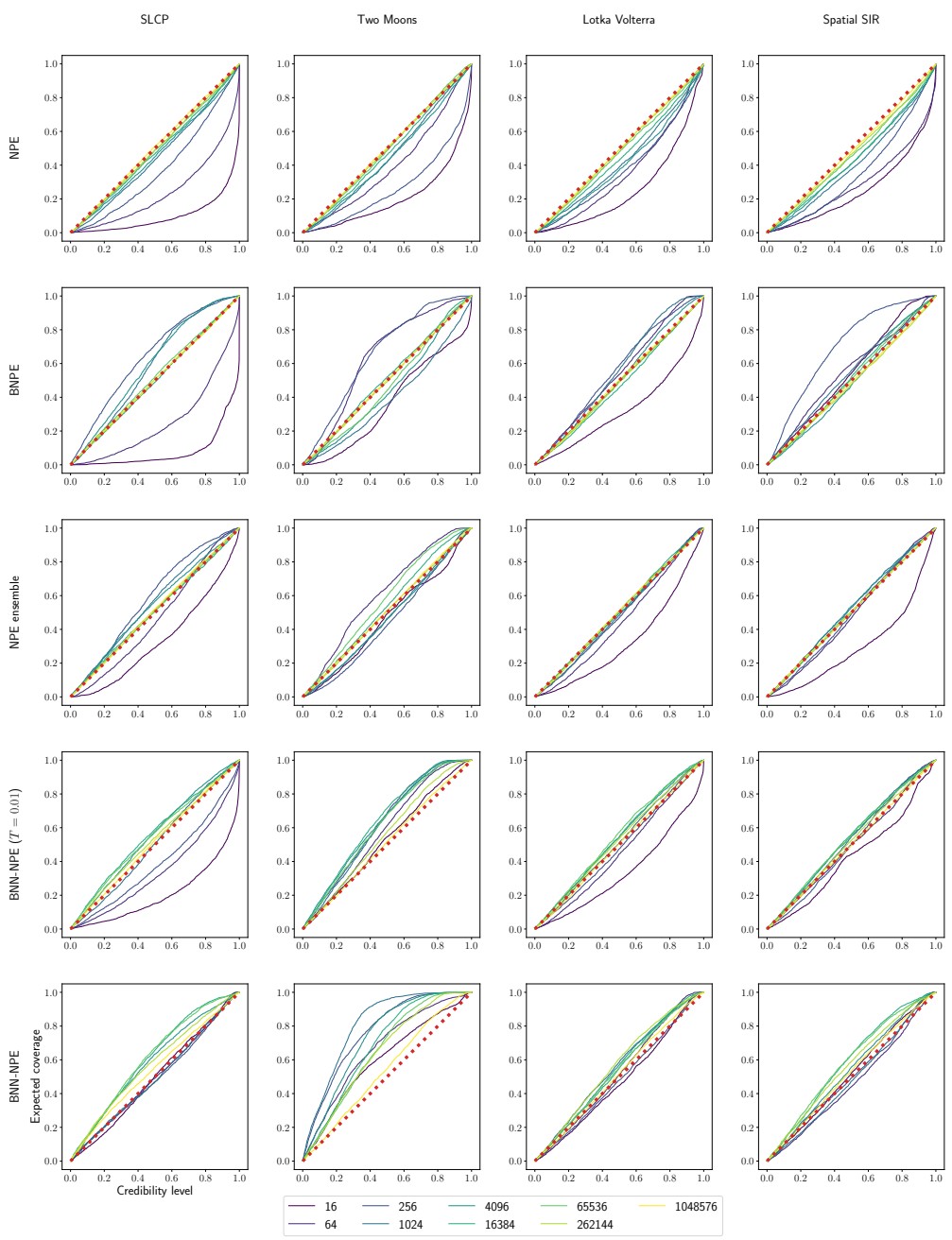

Figure 7: Coverage of different NPE simulation-based inference methods. A calibrated posterior approximation exhibits a coverage AUC of 0. A coverage curve above the diagonal indicates conservativeness and a curve below the diagonal indicates overconfidence. 3 runs are performed, and the median is reported.

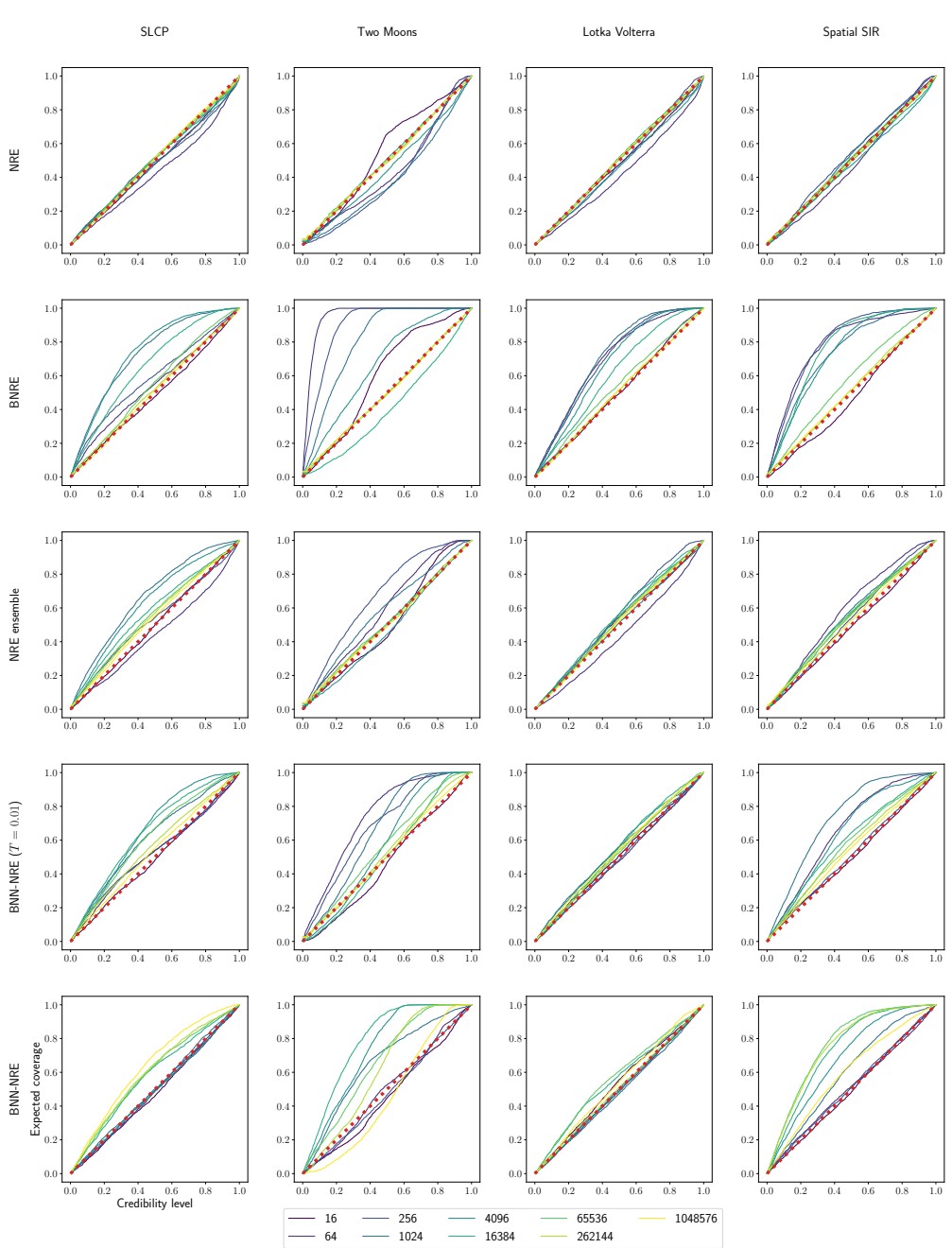

Figure 8: Coverage of different NRE simulation-based inference methods. A calibrated posterior approximation exhibits a coverage AUC of 0. A coverage curve above the diagonal indicates conservativeness and a curve below the diagonal indicates overconfidence. 3 runs are performed, and the median is reported.

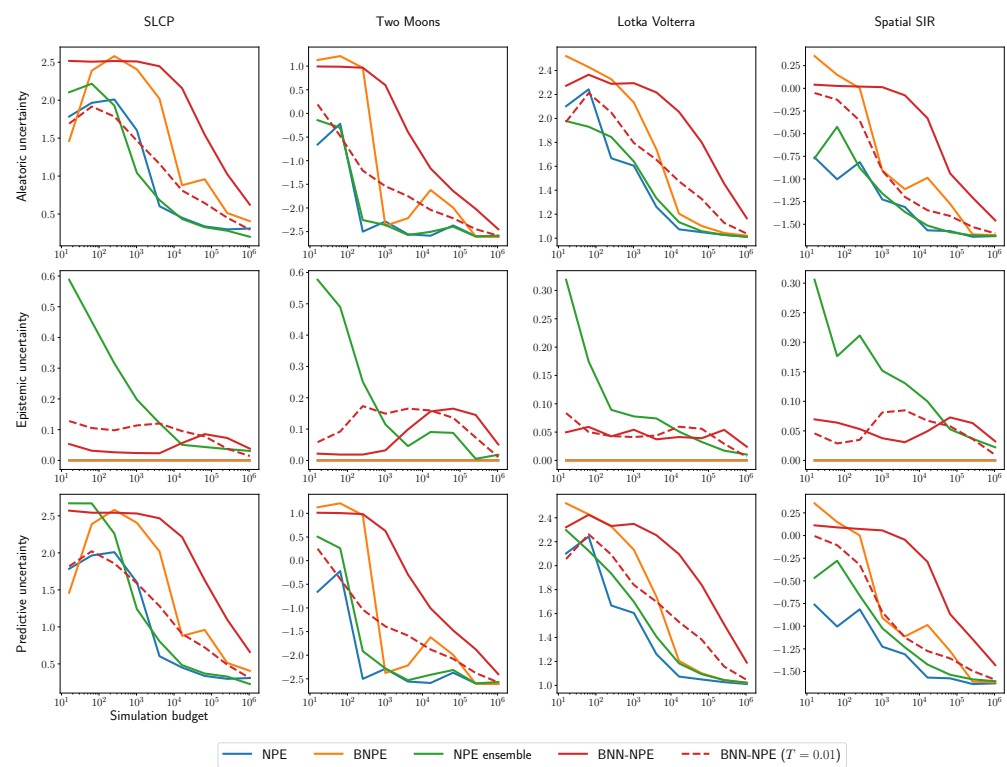

Figure 9: Quantification of the different forms of uncertainties captured by the different NPE-based methods. 3 runs are performed, and the median is reported.

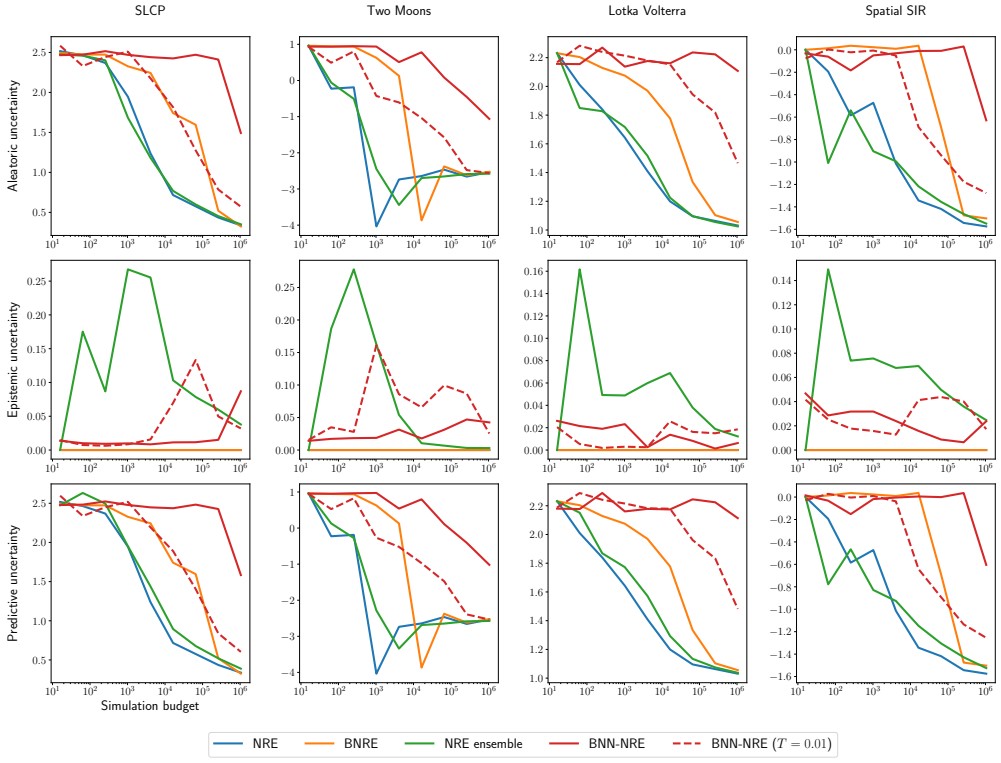

Figure 10: Quantification of the different forms of uncertainties captured by the different NRE-based methods. 3 runs are performed, and the median is reported.

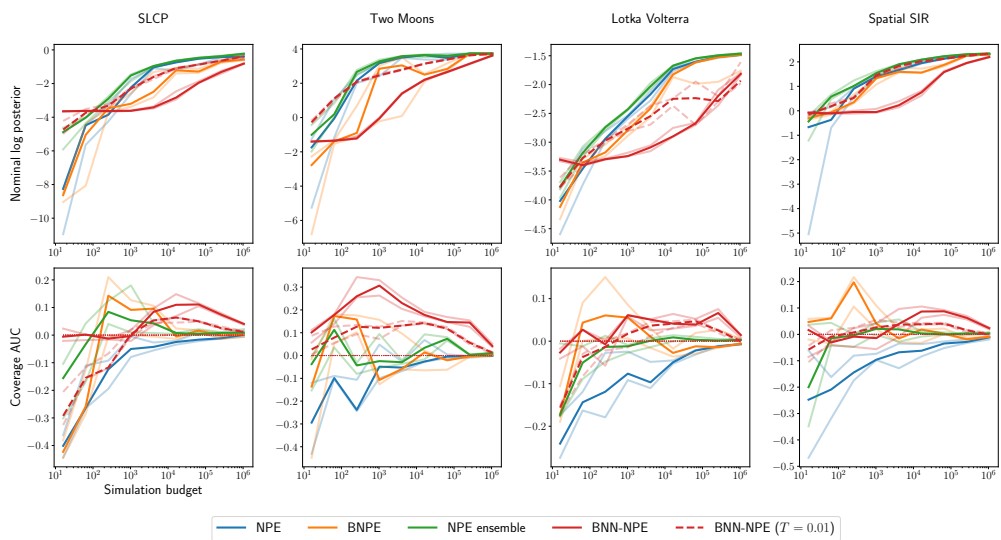

Figure 11: Comparison of different NPE simulation-based inference methods through the nominal log probability and coverage area under the curve. The higher the nominal log probability, the more performant the method is. A calibrated posterior approximation exhibits a coverage AUC of 0. A positive coverage AUC indicates conservativeness, and a negative coverage AUC indicates overconfidence. 3 runs are performed. The median run is reported in plain, and the shaded lines represent the other two runs.

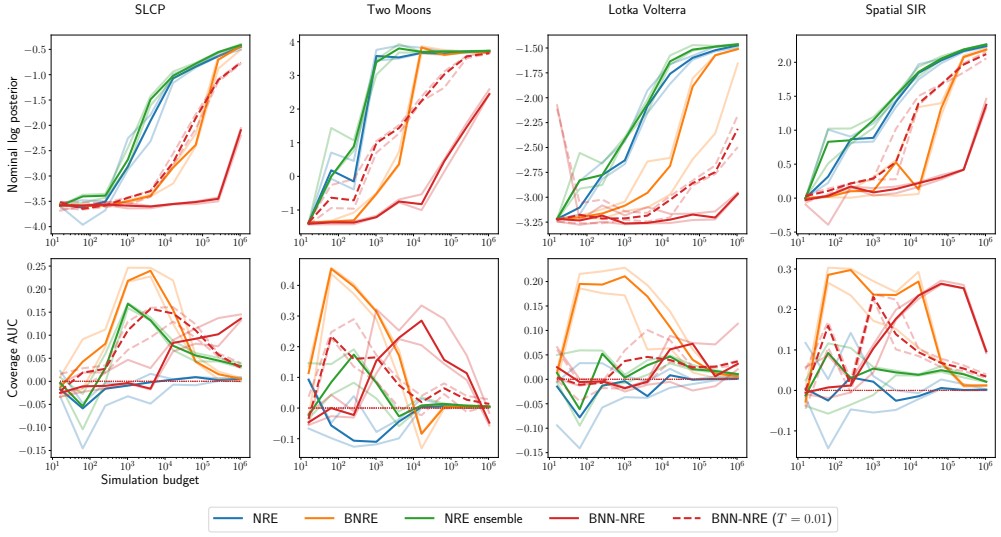

Figure 12: Comparison of different NRE simulation-based inference methods through the nominal log probability and coverage area under the curve. The higher the nominal log probability, the more performant the method is. A calibrated posterior approximation exhibits a coverage AUC of 0. A positive coverage AUC indicates conservativeness, and a negative coverage AUC indicates overconfidence. 3 runs are performed. The median run is reported in plain, and the shaded lines represent the other two runs.

