# OpenReview forum: "Low-Budget Simulation-Based Inference with Bayesian Neural Networks"
_ICLR.cc/2025/Conference — Submitted to ICLR 2025_

### Official Review · Reviewer_WoUh · 2024-10-16

**Soundness:** 2
**Presentation:** 2
**Contribution:** 3
**Rating:** 5
**Confidence:** 2

**Summary:**

The authors wish to provide a method for simulation-based inference for expensive simulators that accounts for the epistemic uncertainty, which tends to be high when available data (simulations) is low. They wish to do so by using a Bayesian neural network in combination with the SBI methods NRE and NPE. They note issue with current priors for Bayesian neural networks in simulation based inference, stating they are not calibrated a priori. They note that using a Gaussian process, with the prior distribution as its mean, can be used to define a prior which is well-calibrated a priori. By optimizing the Bayesian networks prior on the weights to approximate the Gaussian process prior, the Bayesian networks prior becomes approximately well-calibrated. They perform experiments, and show this often leads to better calibrated posteriors in the low-data (low simulation budget) regime.

**Strengths:**

Being able to perform reliable simulation-based inference with fewer simulations would be broadly useful to the scientific community. The idea to use a prior over the weights of a BNN that is compatible with the simulators prior distribution is sensible, and using a Gaussian process to achieve this is to the best of my knowledge novel and clever. The paper is well structured overall (except for where noted below) and includes some interesting experiments, using a good choice of metrics assessing reliability of the posteriors.

**Weaknesses:**

Structure/Presentation:
- Lack of method summary: There's no clear summary of the method, e.g. in the contributions section, an algorithm, or the start of section 3. A brief overview of 3-4 sentences would be very beneficial for readers. It's only towards the end of section 3.2, that the overarching method begins to come together on a first read. In my opinion, the abstract also does not include enough information on the method.
- Lack of clear definitions: The concept of an "a priori-calibrated Bayesian model" should be defined more clearly when it is first introduced.  I presume that an example of a well calibrated a priori model, would be one for which the Bayesian model average at initialization is equal to the simulator parameters prior $p(\theta)$ for any $x \in \mathcal{X}$? Presumably this includes a large variety of useful and non-useful models for modelling epistemic uncertainty (as suggested in equation 8).
- Figures: The credibility figure in Figure 1 is confusing, either the standard deviations are not in order, or there is a typo one of the standard deviations. Figure 3 should have a labeled legend. Many of the font sizes are too small.

Experiments:
- Limited experimental runs: Only 3 runs are performed for each experiment, and only the median is reported. This makes it hard to assess if differences between methods are significant. Repeating with more runs would be beneficial, although I am sympathetic to limitations in computational budget.
- The BNN-NRE method appears to perform worse than NRE in terms of mass placed on true parameters, even for low simulation budgets (100-1000 in Figure 2), which is the domain where the method is proposed to be beneficial. Similarly, the results for the NPE case are not particularly convincing in terms of the mass placed on the true parameters. The coverage properties do appear to have improved, but coverage alone is not indicative of a good posterior estimate.
- The results don't align with an intuitive understanding of epistemic uncertainty. For example, in figure 3, in NPE, we can see 4000 simulations produces a reasonable posterior estimate, whereas BNN-NPE is still massively conservative, even with 65536 simulations, suggesting overestimation of epistemic uncertainty. This suggests simple alternative approaches such as training posterior estimates on subsets of the data with standard methods, such as NPE, and mixing the resulting posteriors, would likely be more effective. I understand the temperature parameter is introduced to limit this problem, but this then introduces another hard to choose hyperparameter (along with the Gaussian process parameters introduced).

**Questions:**

How can we be sure that any benefits are from the BNN modelling epistemic uncertainty, and not the altered initialization? For example, if we "pretrained" NPE/NRE to prior samples (and $x$ simulated from some noise distribution), such that at "initialization" the posterior estimate would be approximately equal to the prior for any $x$, it would be interesting to see if there is still be any benefit to the introduced method.

Most the benefits seem to be in the very small data regime (<100 simulations). I am not convinced that this scenario is of particular interest the scientific community, could you give an example of a practical case where this is useful?

Why is the convergence to the NPE solution so slow in terms of the number of simulations? Can the prior be altered to avoid this problem, rather than introducing a temperature parameter?

---

> ### Author Response · Authors · 2024-11-14
> **Rebuttal**
>
> First, thank you for your review. We are happy to hear that you found the contribution broadly useful to the scientific community, sensible, novel and clever, that the paper mostly well-structured and that the experiments interesting and that the choice of metrics is appropriate.
>
> We adress your concerns bellow.
>
> > Lack of method summary: There's no clear summary of the method, e.g. in the contributions section, an algorithm, or the start of section 3. A brief overview of 3-4 sentences would be very beneficial for readers. It's only towards the end of section 3.2, that the overarching method begins to come together on a first read. In my opinion, the abstract also does not include enough information on the method.
>
> Thanks for pointing this out. We have added the following sentence at the beginning of section 3 to guide the reader.
> "In this section, we propose a novel SBI algorithm based on Bayesian neural networks with tuned prior on weights. The algorithm first optimizes a prior on weight to have desirable conservativeness properties. This tuned prior is then used to compute an approximate posterior on weights that is itself used to make predictions through Bayesian model averaging."
>
> > Lack of clear definitions: The concept of an "a priori-calibrated Bayesian model" should be defined more clearly when it is first introduced. I presume that an example of a well calibrated a priori model, would be one for which the Bayesian model average at initialization is equal to the simulator parameters prior $p(\theta)$ for any $x \ in \mathcal{X}$? Presumably this includes a large variety of useful and non-useful models for modelling epistemic uncertainty (as suggested in equation 8).
>
> An a-priori calibrated Bayesian model is defined as a prior over weights that lead to a calibrated Bayesian model average. If this Bayesian model average is the prior $p(\theta)$ for all $x$, this is indeed a calibrated model. We added the following sentence to make things clearer:
> "This means that the Bayesian model average computed using the prior normal on weights $p(w)$ is not calibrated."
>
> > Figures: The credibility figure in Figure 1 is confusing, either the standard deviations are not in order, or there is a typo one of the standard deviations. Figure 3 should have a labeled legend. Many of the font sizes are too small.
>
> Indeed, there are two standard deviations that are not in order, this has been changed. We also added the meaning of the legend of Figure 3 in the caption.
>
> > Limited experimental runs: Only 3 runs are performed for each experiment, and only the median is reported. This makes it hard to assess if differences between methods are significant. Repeating with more runs would be beneficial, although I am sympathetic to limitations in computational budget.
>
> We agree that more runs would have been preferable. However, we estimated the computational cost of 3 runs to be approximately 25,000 GPU hours. This is because each point in the reported curves corresponds to 3 full training and testing procedures, and testing with Bayesian neural networks involves the approximation of the Bayesian model average for each test sample. Unfortunately, we do not have the computational power to do many more runs. That being said, we see from Figure 2 that all the methods but BNN-NPE show a coverage AUC below -0.1 for low simulation budgets on at least one benchmark. BNN-NPE shows only positive or very slightly negative coverage AUC. We believe we can then conclude that using BNN improves the conservativeness even if we have only 3 runs.

---

> > ### Author Response · Authors · 2024-11-14
> > **Rebuttal**
> >
> > > The BNN-NRE method appears to perform worse than NRE in terms of mass placed on true parameters, even for low simulation budgets (100-1000 in Figure 2), which is the domain where the method is proposed to be beneficial. Similarly, the results for the NPE case are not particularly convincing in terms of the mass placed on the true parameters. The coverage properties do appear to have improved, but coverage alone is not indicative of a good posterior estimate.
> >
> > Indeed, this is a limitation that our method might perform worse in terms of mass placed on true parameters, but this is not what we aim for. This work follows the philosophy highlighted by Hermans et al., 2022, where they argue that to be useful for scientific application, posterior approximations must be conservative before anything else. If that is not the case, then they cannot be used reliably for downstream scientific tasks. In this work, the main objective is to build SBI methods that are conservative for low simulation budgets. In that regard, our method provides significant improvement. If one is only interested in placing the mass on true parameters for their application and not in conservativeness, then our method is probably not the method to use.
> >
> > Joeri Hermans, Arnaud Delaunoy, Francois Rozet, Antoine Wehenkel, Volodimir Begy, and Gilles Louppe. A crisis in simulation-based inference? beware, your posterior approximations can be unfaithful. Transactions on Machine Learning Research, 2022
> >
> > > The results don't align with an intuitive understanding of epistemic uncertainty. For example, in figure 3, in NPE, we can see 4000 simulations produces a reasonable posterior estimate, whereas BNN-NPE is still massively conservative, even with 65536 simulations, suggesting overestimation of epistemic uncertainty. This suggests simple alternative approaches such as training posterior estimates on subsets of the data with standard methods, such as NPE, and mixing the resulting posteriors, would likely be more effective. I understand the temperature parameter is introduced to limit this problem, but this then introduces another hard to choose hyperparameter (along with the Gaussian process parameters introduced).
> >
> > If the model is very uncertain about where to put the mass, then providing a large credible interval means that epistemic uncertainty has been correctly estimated. A good estimation of epistemic uncertainty does not mean providing the correct posterior. In opposition, for a budget of 256, we see that NPE provides a smaller interval than BNN-NPE, but the interval produced by NPE is completely wrong. This means that BNN-NPE estimated the epistemic uncertainty correctly by outputting a very uncertain posterior approximation, while NPE outputs an overconfident posterior approximation. We should also keep in mind that Figure 3 only provides an example, and the correct estimation of epistemic uncertainty cannot be grasped from a single example. To have a correct view of how well epistemic uncertainty is captured, one should look at the coverage curves.
> >
> > > How can we be sure that any benefits are from the BNN modelling epistemic uncertainty, and not the altered initialization? For example, if we "pretrained" NPE/NRE to prior samples (and
> >  simulated from some noise distribution), such that at "initialization" the posterior estimate would be approximately equal to the prior for any, it would be interesting to see if there is still be any benefit to the introduced method.
> >
> > The benefits sure come first from the fact that we have an appropriate prior over weights, which is the main contribution. Then, either the maximum a posteriori could be used or the full Bayesian posterior over weights. In both settings, we benefit from having a better prior, but to capture epistemic uncertainty, using the full Bayesian posterior is needed.
> >
> > > Most the benefits seem to be in the very small data regime (<100 simulations). I am not convinced that this scenario is of particular interest the scientific community, could you give an example of a practical case where this is useful?
> >
> > First, it should be noted that what happens for <100 simulations on our benchmarks could happen for more simulations on much more complicated simulators. Solving the issue for very low budgets on our benchmarks is hence desired. Second, the cosmological application has been chosen to be one of such examples. 35 million CPU hours have been dedicated to generating $44,000$ simulations. This is roughly equivalent to 33 CPU-days for **each single** simulation. This is not something manageable without having access to large computing facilities. Before that massive amount of computing was dedicated to those simulations, a too low amount of simulations were available. With our method, it would have been possible to produce calibrated approximations with a reasonable amount of computation.

---

> ### Author Response · Authors · 2024-11-14
> **Rebuttal**
>
> > Why is the convergence to the NPE solution so slow in terms of the number of simulations? Can the prior be altered to avoid this problem, rather than introducing a temperature parameter?
>
> The prior we define is the reason why we obtain calibrated approximations. Altering the prior would then probably negatively affect the coverage results, as seen in Figure 1. Using a temperature that is too low could also affect the coverage results; it should be tuned to have the appropriate trade-off.
>
> ---
>
> Overall, we think we have addressed each concern raised in your review and implemented changes accordingly. While these improvements appear to resolve the identified issues, we'd value your perspective on whether any aspects still fall short of expectations. If the changes adequately address your concerns, we'd appreciate having this reflected in the evaluation.

---

> > ### Comment · Reviewer_WoUh · 2024-11-17
> >
> > Thanks you for the response and addressing most of my concerns, unfortunately a few still remain. I would argue that the average mass placed on the true parameters is a reasonable aim even when conservative inference is desirable, due to it's close relation to the forward KL divergence $KL(p^*||q)$, which is generally considered a mass covering objective. Or conversely, I would expect a model underestimating epistemic uncertainty to perform poorly by that metric (i,.e. occasionally having posteriors that place little mass on the true parameter).
> >
> > When we are discussing very low data regimes (<100), the fitting procedure will become very influential, i.e. size of validation set and initialisation. I see in the code a validation set has been used which is nice to see (maybe should be mentioned in the paper), however, as mentioned in my previous comment, I wonder if pretraining NPE/NRE to prior samples would make any performance differences go away, i.e. by providing a better initialisation.

---

> > > ### Author Response · Authors · 2024-11-18
> > >
> > > Thank you for acknowledging the improvements regarding most of your concerns, which were the lack of method summary, the lack of clear definition, confusing figures, the limited number of runs, and the clarification of epistemic uncertainty. While we answer the few remaining concerns, below we would like to invite you to already reflect those changes in your evaluation, which you can do by editing the review.
> > >
> > > Regarding the importance of having the average mass placed at the right place, we agree that having a low KL divergence is something desirable and that if we consider only standard methods, a low KL divergence will usually be correlated with good coverage. However, in all generality, for non-null KL divergence values, it is possible for approximation A to have lower KL than approximation B, but approximation B is conservative while approximation A is not (see Figure 1 in Hermans et al., 2022). Following the arguments of Hermans et al., 2022, approximation B would, in this case, still be preferred for some scientific applications as non conservative methods cannot provide reliable results. We develop our method for this setting, where conservativeness is even more important than placing the mass at the right place, even if both are important. Our empirical results suggest that our method is indeed conservative while others are not. Although it would indeed be better to, on top of that, not lose anything in terms of KL divergence with the true posterior, conservativeness is an improvement. In conclusion, while we agree that our method is not perfect and we acknowledge that in the conclusion section, this is still an improvement over existing methods which can dangerously overfit in the low budget regime.
> > >
> > > We added a mention of the use of a validation set in Appendix C. Regarding the suggestion of a change of initialization for NRE and NPE, indeed, the prior we develop might find many other applications, either in terms of initialization or it can also be used as prior without training a BNN. If we had to cover all the possible use cases, it would take far more than 9 pages and would dilute the main message. On top of that, our goal is both to introduce a good prior and to capture computational uncertainty, which is only possible with BNNs. We believe that those suggestions would be nice to publish in subsequent papers, but we do not see this as a drawback of our method, and we hope it will not be considered as such.

---

> > > > ### Author Response · Authors · 2024-11-28
> > > >
> > > > Dear reviewer,
> > > >
> > > > Thanks for initiating the discussion. Did our response improve your view of the paper? If there are still points to clarify we would be glad to use the last few days to discuss those.

---

### Official Review · Reviewer_9hZV · 2024-10-25

**Soundness:** 3
**Presentation:** 2
**Contribution:** 3
**Rating:** 6
**Confidence:** 5

**Summary:**

The authors aim to produce well-calibrated posterior in simulation-based inference, in particular in cases where the simulator is very expensive and, therefore, few simulations can be used as training data. To achieve this, the authors propose to use Bayesian neural networks. They develop a novel prior which leads to a-priori well-calibrated posteriors. They apply their method to several toy simulators and to a physics simulator.

**Strengths:**

The idea to generate neural network weight priors which lead to a-priori well-calibrated posteriors is novel, interesting, and potentially impactful. The methodology which the authors employ to achieve this is novel, elegant, and rigorous. I thoroughly enjoyed reading these parts of the paper. In addition, the authors demonstrate that the method can be applied across methods (NPE & NRE) and they evaluate the method on a series of useful tasks.

**Weaknesses:**

(1) The paper overstates its claims.
My main issue with this paper is that it overstates its claims and does not acknowledge the weakness of empirical results. Looking at Figure 2, and in particular for NRE: BNN-NRE _never_ reaches the log posterior of the other methods (even for 1M simulations). Yet the authors state that `the nominal log posterior density is on par with other methods for very high simulation budgets`. Where does this claim come from. Similarly, in Figure 2 (NRE), the authors do not acknowledge that the AUC is not particularly much higher for BNN-NRE as compared to NRE, even for 10 simulations. Indeed, for three of the four tasks NRE has a higher coverage AUC than BNN-NRE for 10 simulations. On top of this, for many tasks and simulation budgets, BNN-NRE does have a negative AUC, but the authors simply claim that they `show positive coverage AUC`. Finally,

(2) The empirical results are weak.
Second, to me, the empirical results are difficult to interpret. (A) Many of the curves shown in Figure 2 are very noisy and irregular. It might be beneficial to average across more seeds to observe clear trends. (B) As a potential user, I would find it very worrisome that BNN-NRE has _significantly_ lower nominal log-posterior than standard methods. Indeed, for some tasks, it seems to require 5 orders of magnitude (for many other tasks 3 orders of magnitude) more simulations than standard methods.

**Questions:**

No questions.

---

> ### Author Response · Authors · 2024-11-14
> **Rebuttal**
>
> First, thank you for your review. We are happy to hear that you found the contribution novel, interesting, potentially impactful, elegant and rigorous.
>
> We adress your concerns bellow.
>
> > The paper overstates its claims. My main issue with this paper is that it overstates its claims and does not acknowledge the weakness of empirical results. Looking at Figure 2, and in particular for NRE: BNN-NRE never reaches the log posterior of the other methods (even for 1M simulations). Yet the authors state that the nominal log posterior density is on par with other methods for very high simulation budgets. Where does this claim come from. Similarly, in Figure 2 (NRE), the authors do not acknowledge that the AUC is not particularly much higher for BNN-NRE as compared to NRE, even for 10 simulations. Indeed, for three of the four tasks NRE has a higher coverage AUC than BNN-NRE for 10 simulations. On top of this, for many tasks and simulation budgets, BNN-NRE does have a negative AUC, but the authors simply claim that they show positive coverage AUC. Finally,
>
> We totally agree that our method has limitations. We have tried to be as transparent as possible regarding those limitations by including a limitation paragraph at the end of the conclusion highlighting all the mentioned limitations. The differences between BNN-NPE and BNN-NRE are also highlighted in lines 416-418. We understand that some sentences might sound like overstatements, and we have now modified those. We have modified the sentence about log posterior density as follows:
> "We also observe that the nominal log posterior density is on par with other methods for very high simulation budgets, on most benchmarks and with an appropriate temperature, but that more samples are required to achieve high values."
> Regarding the coverage AUC of NRE and BNN-NRE, in our opinion, BNN-NRE has better coverage values. We observe that NRE reaches coverage AUC values of -0.05 on SLCP, -0.1 on Two Moons, almost -0.1 on Lotka-Volterra, and almost -0.05 on Statial SIR. In opposition, BNN-NRE shows coverage AUC that is at worse, around -0.03, and is much more often positive. The coverage curves of BNN-NRE are above NRE at least 90\% of the time. We agree that stating that BNN-NRE always shows positive coverage is, strictly speaking, wrong. We have updated the claim as follows:
> "Figure 2 compares simulation-based inference methods with and without accounting for computational uncertainty. We observe that BNNs equipped with our prior and without temperature show positive, or only slightly negative, coverage AUC even for simulation budgets as low as $O(10)$."
>
> > The empirical results are weak.
>
> We agree that our method shows empirical limitations as it requires higher simulation budgets to reach similar log posterior density. However, the empirical results clearly show that using Bayesian neural networks with our prior leads to better coverage AUC. All other methods are shown to have dangerously low coverage AUC in low simulation budget regime. We lose in statistical power to avoid risking false inference, which is the main objective of this work.
>
> > Second, to me, the empirical results are difficult to interpret. (A) Many of the curves shown in Figure 2 are very noisy and irregular. It might be beneficial to average across more seeds to observe clear trends.
>
> We agree that more runs would have been preferable. However, we estimated the computational cost of 3 runs to be approximately 25,000 GPU hours. This is because each point in the reported curves corresponds to 3 full training and testing procedures, and testing with Bayesian neural networks involves the approximation of the Bayesian model average for each test sample. Unfortunately, we do not have the computational power to do many more runs. That being said, as mentioned above, even if the curves are a bit noisy, they are sufficient to validate the fact that using Bayesian neural networks with our prior leads to better coverage AUC.

---

> ### Author Response · Authors · 2024-11-14
> **Rebuttal**
>
> > As a potential user, I would find it very worrisome that BNN-NRE has significantly lower nominal log-posterior than standard methods. Indeed, for some tasks, it seems to require 5 orders of magnitude (for many other tasks 3 orders of magnitude) more simulations than standard methods.
>
> It depends on what kind of application one is working on and what we are trying to achieve. This work follows the philosophy highlighted by Hermans et al., 2022, where they argue that to be useful for scientific application, posterior approximations must be conservative before anything else. If that is not the case, then they cannot be used reliably for downstream scientific tasks. In this work, the main objective is to build SBI methods that are conservative for low simulation budgets. In some settings, people might indeed not care that much about conservativeness and focus on statistical power. Our work does not aim to address this setting, and in those cases, other methods might indeed be preferred. We would also like to highlight that we focus on conservative simulation-based inference in the low simulation budget regime, a regime where other methods fall short. For large datasets, others methods like standard or balanced NPE/NRE might be prefered.
>
> Joeri Hermans, Arnaud Delaunoy, Francois Rozet, Antoine Wehenkel, Volodimir Begy, and Gilles Louppe. A crisis in simulation-based inference? beware, your posterior approximations can be unfaithful. Transactions on Machine Learning Research, 2022
>
> ---
>
> Overall, we think we have addressed each concern raised in your review and implemented changes accordingly. While these improvements appear to resolve the identified issues, we'd value your perspective on whether any aspects still fall short of expectations. If the changes adequately address your concerns, we'd appreciate having this reflected in the evaluation.

---

> > ### Comment · Reviewer_9hZV · 2024-11-18
> > **Response**
> >
> > Thank you for the response and for the modifications. Unfortunately, I think that the updated manuscript (and in particular the updated sentence) still contains overstatements and a lack of clearly described limitations. First:
> >
> > > We also observe that the nominal log posterior density is on par with other methods for very high simulation budgets, on most benchmarks and with an appropriate temperature, but that more samples are required to achieve high values.
> >
> > Looking at Figure 2, this is simply not true. Across 8 benchmarks (4 for NPE, 4 for NRE), the BNN (with T=0.01) does **not** reach the same nominal log posterior as NPE/NRE: LV for NPE, and SLCP/LV/SIR for NRE. Four of eight tasks is not "most benchmarks".
> >
> > Second, while the authors do state limitations, I think the extent of underconfidence of the method is never explicitly discussed. The authors state that their method
> >
> > > might need more simulated data (L514)
> >
> > but, to me, the real limitation is that the BNN requires **multiple orders of magnitude** more simulations than non-BNN versions.
> >
> > Finally, regarding BNN-NRE: I still think that calling BNN-NRE "reliable" is not warranted. BNN-NRE has negative AUC on multiple tasks: 5/10 on SLCP, on 3/10 for 2M, on 4/10 for LV, and 1/10 for SIR. Yes, it is less overconfident than other methods, but I still think that this behavior deserves a clear description in the limitations. Yet, it is never mentioned in the limitations in the conclusion and it is also not explicitly described anywhere else. IMO the passing reference "or slightly negative" attempts to sweep the limitations under the rug.

---

> > > ### Author Response · Authors · 2024-11-18
> > >
> > > Thanks for your response. We understand your concerns and have further update the manuscript to make the limitations clearer. In particular, we have modified the problematic paragraph as follows:
> > >
> > > ```
> > > Figure 2 compares simulation-based inference methods with and without accounting for computational uncertainty. We observe that BNNs equipped with our prior and without temperature show positive, or only slightly negative, coverage AUC even for simulation budgets as low as $O(10)$. Negative coverage AUC is still observed, and hence conservativeness is not strictly guaranteed. However, this constitutes a significant improvement over the other method in that regard. The coverage curves are reported in Appendix D. We conclude that BNNs can hence be more reliably used than the other benchmarked methods when the simulator is expensive and few simulations are available. We observe that increasing the reliability comes with the drawback of requiring more simulations than the other methods to reach similar nominal log posterior density values. Without temperature, a few orders of magnitude more samples might be needed. However, in theory, as the amount of sample increases, the effect of the prior diminishes, and BNNs should reach the same nominal log posterior density as standard methods. By adding a temperature to the prior, its effect is diminished and better nominal log posterior density values are observed. From these observations, general guidelines to set the temperature include increasing $T$ if overconfidence is observed and decreasing it if low predictive performance is observed.
> > > ```
> > >
> > > We have also further updated the limitations paragraph as follows:
> > >
> > > ```
> > > Using BNNs for simulation-based inference also comes with limitations. The first observed limitation is that the Bayesian neural network based methods might need orders of magnitude more simulated data in order to reach a predictive power similar to methods that do not use BNNs, such as NPE. While we showed that this limitation can be mitigated by tuning the temperature appropriately, this is something that might require trials and errors. A second limitation is the computational cost of predictions. When training a BNN using variational inference, the training cost remains on a similar scale as standard neural networks. At prediction time, however, the Bayesian model average described in Equation 4 must be approximated, and this requires a neural network evaluation for each Monte Carlo sample in the approximation. The computational cost of predictions then scales linearly with the number $M$ of Monte Carlo samples. Finally, although our method significantly improves the reliability over standard methods for low simulation budgets, conservativeness is not strictly guaranteed. There are no theoretical guarantees and negative coverage AUC may still be observed.
> > > ```
> > >
> > > In the abstract, we have also replaced the sentence
> > >
> > > ```
> > > We design a family of Bayesian neural network priors that are tailored for inference and show that they lead to well-calibrated posteriors on tested benchmarks, even when as few as $O(10)$ simulations are available."
> > > ```
> > > by
> > > ```
> > > We design a family of Bayesian neural network priors that are tailored for inference and show that they lead to better calibrated posteriors than standard methods on tested benchmarks, even when as few as $O(10)$ simulations are available."
> > > ```
> > > The PDF has been updated to include those new changes. We hope that those paragraphs are now a better reflection of our work. We remain committed to making limitations clearer in case further modifications are needed.

---

> > > > ### Comment · Reviewer_9hZV · 2024-11-19
> > > > **Response**
> > > >
> > > > Thank you for the quick response! The updated statements align with the results in the paper.
> > > >
> > > > As I had written in my initial response, I believe that the presented method is elegant and novel. In addition, I think that the results (in particular for NPE) demonstrate SOTA if one aims to achieve conservative posteriors.
> > > >
> > > > IMO, the major limitations are still that (1) the method requires orders of magnitude more simulations than "standard" NPE, and (2) that the results are relatively noisy and hard to interpret due to only three seeds (I think the computational budget could be remedied by using fewer sets of #sims: Ten different sets seems excessive to me).
> > > >
> > > > Balancing these advantages and limitations, I am leaning towards acceptance and I am increasing my score to 6.

---

> > > > > ### Author Response · Authors · 2024-11-19
> > > > >
> > > > > We are glad to hear that the limitations are now appropriately stated. Thanks for engaging constructively in the discussion and for updating your evaluation!

---

### Official Review · Reviewer_VQN7 · 2024-11-01

**Soundness:** 3
**Presentation:** 2
**Contribution:** 3
**Rating:** 6
**Confidence:** 4

**Summary:**

The authors propose a novel approach for simulation-based inference for low-budget problems using Bayesian neural networks. They additionally propose to use tailored priors for the neural network weights which generally leads to more conservative posterior estimates. They evaluate their method on several experimental models against multiple baselines demonstrating promising results for SBI when the computational budget is limited.

**Strengths:**

- The paper proposes a novel contribution for simulation-based inference. The method is in my opinion very relevant to the community and interesting as a low-budget solution for inferential problems.
- The motivation and derivation of the tailored prior is in my opinion particularly well done and convincing.
- Empirically, the method seems to achieve the motivated goal: having calibrated posteriors for low simulation budgets.
- The paper is well written and easy to follow.

**Weaknesses:**

The evaluations and presentations of the results could in my opinion be improved.

- The authors evaluate their method with the expected coverage (EC). In Figure 2, it is in my opinion very difficult to draw conclusions which methods works best. This is likely due to the fact that only 3 runs have been evaluated. Given the complexity of SBI and the high variance of the inferential results, I think they should do at least 10 evaluations and report these.
- The authors use as a second evaluation metric the expected posterior log density (EPLD). It is not clear what a good value should look like here or even if a high log density should be desired. When the goal is to have a conservative posterior, is a extremely high EPLD a good measure? The authors should, despite possible drawbacks, have used some divergence to the true posterior, such as MMD, in addition to the other two and report this.
- It is not clear how crucial hyperparameters such as the temperature $T$ or the covariance functions of the reference GP are chosen in practice.
- The authors propose to use cold posteriors, e.g., using a temperature of $T=0.01$. The motivation of this is not clear, given that the entire idea is to use a prior that enforces calibration. Why would it be desirable to in fact reduce the impact of the prior?

#### Minor
- Figures 2 should show the standard errors.
- Figure 3 should show the reference posterior and not only the true parameter value.

**Questions:**

- The results in Figure 3 seem to indicate that NPE is indeed outperforming BNN-NPE here. Would the authors argue that BNN-NPE is preferable over NPE for this task?
- What do the numbers in the legend in Figure 3 mean?

---

> ### Author Response · Authors · 2024-11-14
> **Rebuttal**
>
> First, thank you for your review. We are happy to hear that you found the contribution novel, interesting, and very relevant to the community, that the motivation and derivation of the tailored prior is particularly well done and convincing, that the method seems to empirically achieve the motivated goal and that the paper is well-written and easy to follow. We address your concerns below.
>
> > The authors evaluate their method with the expected coverage (EC). In Figure 2, it is in my opinion very difficult to draw conclusions which methods works best. This is likely due to the fact that only 3 runs have been evaluated. Given the complexity of SBI and the high variance of the inferential results, I think they should do at least 10 evaluations and report these.
>
> We agree that more runs would have been preferable. However, we estimated the computational cost of 3 runs to be approximately 25,000 GPU hours. This is due to the fact that each point in the reported curves corresponds to 3 full training and testing procedures and that testing with Bayesian neural networks involves the approximation of the Bayesian model average for each test sample. Unfortunately, we do not have the computational power to do many more runs. That being said, we see from Figure 2 that all the methods but BNN-NPE show a coverage AUC below -0.1 for low simulation budgets on at least one benchmark. BNN-NPE shows only positive or very slightly negative coverage AUC. We believe we can then conclude that using BNN improves the conservativeness even if we have only 3 runs.
>
> > The authors use as a second evaluation metric the expected posterior log density (EPLD). It is not clear what a good value should look like here or even if a high log density should be desired.
> is to have a conservative posterior, is a extremely high EPLD a good measure? The authors should, despite possible drawbacks, have used some divergence to the true posterior, such as MMD, in addition to the other two and report this.
>
> To evaluate the conservativeness of a posterior approximation, coverage is the only relevant measure. We reported EPLD to compare the methods as to how close they are to the real posterior. EPLD is actually the training objective of NPE methods and can be derived from the expected KL divergence between the approximate posterior and true posterior.
>
> $ w^* = \arg \min_w \mathbb{E}_{p(x)} \left[ KL\left[ p(\theta | x) || \hat{p}_w(\theta | x) \right] \right]$
>
> $ w^* = \arg \min_w \mathbb{E}_{p(\theta, x)} \left[ \frac{p(\theta | x) }{\hat{p}_w(\theta | x)}\right]$
>
> $ w^* = \arg \max_w \mathbb{E}_{p(\theta, x)} \left[\hat{p}_w(\theta | x)\right]$
>
> We indeed do not know what a good EPLD value is, but we know that a method with high EPLD is closer to the real posterior than one with low EPLD, according to the KL divergence.
>
> > It is not clear how crucial hyperparameters such as the temperature T or the covariance functions of the reference GP are chosen in practice.
>
> If you are referring to the temperature $T$ applied on the prior during the posterior approximation, we have given guidelines on lines 408-409. If you are referring to the parameters variance and lengthscale of the kernel, those are indeed hard to choose. The lengthscale is chosen to be a quantile of observed distances when taking two samples from the measurement set. The variance should be guided by domain knowledge as to how close the posterior is expected to be from the prior. In this setting, the priors are uniform and the prior density is then equal on all the parameter space. We have set the standard deviation to half that density value but more refined choices could be made with domain knowledge. This is explained in lines 686-692 in Appendix A.
>
> > The authors propose to use cold posteriors, e.g., using a temperature of T=0.01. The motivation of this is not clear, given that the entire idea is to use a prior that enforces calibration. Why would it be desirable to in fact reduce the impact of the prior?
>
> Indeed, this is counterintuitive. To obtain conservative approximations, it is better not to use a temperature. However, we have observed that in some cases, we were not able to reach satisfactory EPLD values and proposed using temperature as a way to set a trade-off between conservativeness and EPLD values. Note that tempering the prior is something that is often done in practice in Bayesian deep learning.
>
> > Figures 2 should show the standard errors.
>
> They are shown in Figure 11. We didn't include those in Figure 2 because it makes the figure hard to read.
>
> > Figure 3 should show the reference posterior and not only the true parameter value.
>
> The reference posterior is unfortunately unknown, but we can assume that NPE with a simulation budget of 1 million samples should not be far from the reference posterior.

---

> ### Author Response · Authors · 2024-11-14
> **Rebuttal**
>
> > The results in Figure 3 seem to indicate that NPE is indeed outperforming BNN-NPE here. Would the authors argue that BNN-NPE is preferable over NPE for this task?
>
> If we have a close look to what happens for a budget of 256, we observe that BNN-NPE takes into account the epistemic uncertainty and provide a large posterior. However, NPE provides a narrower posterior with its mass centered at the wrong place. NPE is then confident and wrong, while BNN-NPE is not overconfident. This behavior of NPE is what we are trying to avoid with BNN-NPE, and it works for that example.
>
> > What do the numbers in the legend in Figure 3 mean?
>
> They are the simulation budgets. We have updated the caption to make it clearer.
>
> ---
>
> Overall, we think we have addressed each concern raised in your review and implemented changes accordingly. While these improvements appear to resolve the identified issues, we'd value your perspective on whether any aspects still fall short of expectations. If the changes adequately address your concerns, we'd appreciate having this reflected in the evaluation.

---

> ### Comment · Reviewer_VQN7 · 2024-11-19
>
> Thank you for addressing my comments.
>
> > We agree that more runs would have been preferable. However, we estimated the computational cost of 3 runs to be
> approximately 25,000 GPU hours.
>
> I understand that and agree that running SBI experiments is very costly. Given that the results of the paper are, however, of empirical nature and the fact that all reviewers seem to agree, I feel like the results are not conclusive enough with 3 replications. As has been pointed out in another review, I believe limiting the simulation budget to $10^4$ in lieu of more experiments, could make the manuscript more convincing. (Further, given that the paper proposes a solution for low-budget SBI, I find the argument that the BNNs need to compute costly model averages a bit antithetic).
>
> After having read the other reviews, I will leave my score as is.

---

> > ### Author Response · Authors · 2024-11-19
> >
> > Thank you for this thoughtful feedback. While we understand your concern about having only 3 runs, we respectfully disagree with the suggestion to limit the simulation budget to 10^4. Here's why:
> >
> > 1. The high simulation budgets are essential to demonstrate our method's full behavior, particularly showing how it transitions from conservative estimates to approaching standard NPE/NRE performance. Truncating at 10^4 would hide important aspects of our method's characteristics.
> > 2. We agree there's an apparent tension between computational training costs and low simulation budgets. However, it's crucial to distinguish between simulation budgets and training budgets. In many scientific applications like our cosmology example, where each simulation takes ~33 CPU-days, reducing the number of expensive simulator runs is the primary concern, even if it comes at the cost of longer training time. The additional training overhead from model averaging is a worthwhile tradeoff if it enables reliable inference from fewer precious simulations.
> > 3. Importantly, our results show significant improvements in calibration across all tested benchmarks in the low-budget regime (O(10)-O(100) simulations), which is our primary contribution. While we agree more runs would strengthen our empirical validation, the consistency of improved calibration across 24 different scenarios (3 runs × 4 benchmarks × 2 SBI methods) provides strong evidence for the effectiveness of our method where it matters most - in the low simulation budget regime.
> >
> > We want to thank you for engaging in discussion and providing constructive feedback.

---

### Official Review · Reviewer_qwfa · 2024-11-04

**Soundness:** 3
**Presentation:** 3
**Contribution:** 4
**Rating:** 5
**Confidence:** 4

**Summary:**

The paper intodruces a new method for simulation-based inference (SBI) that uses
Bayesian neural networks (BNN) to better account for uncertainty in posterior estimates
due to limited training data. The authors demonstrate that previous attempts of
combining BNN with SBI showed limited success due to the choice of prior on the BNN
weights and they propose an alternative prior more suitable for the SBI setting. To
evaluate the proposed method, the authors show on four benchmarking tasks and on a SBI
use-case from cosmology that the resulting SBI posterior estimates are well-calibrated
even in the low-data regime.

**Strengths:**

### Originality

The idea of using BNN for SBI is not new (as discussed in the paper). However, showing
why previous approaches did not work so well and the proposal of a new type of BNN prior
more suitable for the SBI setting is a valuable contribution. Most of the previous work
on better uncertainty quantification in SBI is discussed adequately in the introduction.
The only paper with a related approach that seems to be missing in the discussion is [Lueckmann et al.
2017, section
2.2](https://proceedings.neurips.cc/paper/2017/hash/addfa9b7e234254d26e9c7f2af1005cb-Abstract.html),
who propose performing SVI on the neural network weights for continual learning across
SNPE rounds. Also, it seems a bit odd to me that the benchmark used are not cited
(except for the spatial SIR by Hermans et al.). The SLCP benchmark was introduced by
Papamakarios et al. (SNLE paper), two moons by Greenberg et al.; and the overall set of
SBI benchmarking tasks was introduced in Lueckmann et al. 2021.

### Quality

The technical contributions of the paper appear as sound and the selected methods for
applying BNN to SBI are appropriate. The experimental results tend to support the
initial claim of obtaining well-calibrated posteriors in low-data regime. However, the
number of performed experiments is quite low (three) and the results appear quite noisy.

In general, it should be made clearer how strong the trade-off is between posterior
calibration and posterior accuracy. At the moment it seems that the BNN approach tends
to be quite underconfident in some scenarios (e.g., BNN-NRE on all benchmarks). I
therefore suggest that more experiments should be run and results should be reported
with error bars (e.g., 5-10 runs with error bars showing the standard error of the mean
performance). Additionally, I think it would be good to also check the posterior
predictive distribution for the cases where the BNN has low nominal log posterior values
even in high-data regimes.

### Clarity

Overall, the paper is well-written and clearly-structured. I have a couple of remarks
that should be addressed to improve the clarity.

- In section 3, especially in section 3.2, it should be explained more clearly why and
  how the prior has to be tuned to be used for the BNN approach. In the classical SBI
  setting, the prior is usually set a priori using expert knowledge. This step usually
  does not involve using simulated data for an optimization procedure. In the BNN
  approach, it seems that the prior has to be tuned with actual simulations (e.g., lines
  254-261). Do these simulations have to be run in addition to the training data? Are
  they accounted for the in the budgets in the benchmarks?
- Figure 1 needs a couple of clarifications. The caption says it's a visualization of
  the tuned prior. Then, the next sentence says the left panels show posterior functions
  sampled from the *tuned prior prior over the neural network's weights*. How are these
  samples obtained and what are thet supposed to show? Are they obtained before or after
  SBI training? Similarly question for the last panel: is the calibration calculated
  after the full SBI training or only after the prior tuning?

In general, it seems that a better explanation of the steps involved in setting up and
training the BNN-NPE (NRE) approach. I suggest adding more details in the next and
adding an algorithm scheme in the text or appendix.

### Significance

Uncertainty quantification in neural SBI methods on the posterior approximation itself
is an important and timely problem. Especially in low-data regimes, common neural SBI
methods like NPE struggle. The proposed BNN approach with a Gaussian process prior
as proposed here appears as an important contribution for addressing this issue.

**Weaknesses:**

I outlined several concerns and questions above. To summarize to most important points:

- the experimental results are difficult to interpret because they show the media of
  only three repetitions. More repetitions and error bars would be better here.
- the high underconfidence of the BNN approach in how-data regimes is concerning.
  Additional evaluation of the posterior predictive distributions would be appropriate.
- the choice and construction of the prior from simulated should be explained more
  clearly. A better explanation ideally will resolve the questions on the general
  procedure and on Figure 1 above.

**Questions:**

1) Do these simulations for prior tuning have to be run in addition to the training data?
2) Are they accounted for the in the budgets in the benchmarks?
3) NPE / NRE ensembles seems to perform similarly well compared to the BNN approach. How
  does it perform on the cosmology example? How does it compare to the BNN approach in
  terms of computational budget? Do you think ensembles could be a good alternative when
  computational budgets are limited? What are the disadvantages of ensembles compared to
  BNN for SBI?
4) More generally, what is the computational complexity of the BNN approach compared to
  NPE (ensembles)?

---

> ### Author Response · Authors · 2024-11-14
> **Rebuttal**
>
> First, thank you for your review. We are happy to hear that you found the contributions valuable and that it addresses a timely problem, that you think that the technical contributions are sound and that the experimental results support the claims, and that you found the paper well-written and clearly structured. We address your concerns below.
>
> > The only paper with a related approach that seems to be missing in the discussion is Lueckmann et al. 2017, section 2.2, who propose performing SVI on the neural network weights for continual learning across SNPE rounds. Also, it seems a bit odd to me that the benchmark used are not cited (except for the spatial SIR by Hermans et al.). The SLCP benchmark was introduced by Papamakarios et al. (SNLE paper), two moons by Greenberg et al.; and the overall set of SBI benchmarking tasks was introduced in Lueckmann et al. 2021.
>
> Thanks for pointing it out. We were not aware of the work by Lueckmann et al. (2017), and we now discuss it in the introduction. The benchmarks are appropriately cited in Appendix B.
>
> >In section 3, especially in section 3.2, it should be explained more clearly why and how the prior has to be tuned to be used for the BNN approach. In the classical SBI setting, the prior is usually set a priori using expert knowledge. This step usually does not involve using simulated data for an optimization procedure. In the BNN approach, it seems that the prior has to be tuned with actual simulations (e.g., lines 254-261). Do these simulations have to be run in addition to the training data? Are they accounted for the in the budgets in the benchmarks?
>
> > Do these simulations for prior tuning have to be run in addition to the training data? Are they accounted for the in the budgets in the benchmarks?
>
> First, we would like to clarify what kind of prior we are talking about. In the classical SBI setting, a prior over the simulator's parameters $p(\theta)$ is set a priori using expert knowledge. From that, an approximate posterior model based on a neural network with fixed weights $w$ is computed $p(\theta | x, w)$. In this paper, we are also using a prior over simulator's parameters $p(\theta)$ that is set using expert knowledge and not learned or optimized in any way. However, we use a Bayesian neural network as an approximate model, and the approximate posterior becomes $\hat{p}(\theta | x) = \int p(\theta | x, w) p(w | D) dw$. The term $p(w | D)$ requires the introduction of a prior $p(w)$ that is, this time, over the neural network's weight and not the simulator's parameters. We optimize this prior to match the behavior of a Gaussian process in the functional domain. We mention in lines 258-265 several ways to construct a measurement set for the optimization. In the experiments, we only use simulations to derive the support of the distribution, and the simulations used are the same as the training simulations. No additional simulations are then used, and all the simulations that are used are taken into account in the simulation budget.
>
> > Figure 1 needs a couple of clarifications. The caption says it's a visualization of the tuned prior. Then, the next sentence says the left panels show posterior functions sampled from the tuned prior prior over the neural network's weights. How are these samples obtained and what are thet supposed to show? Are they obtained before or after SBI training? Similarly question for the last panel: is the calibration calculated after the full SBI training or only after the prior tuning?
>
> Again, the confusion might come from the fact that there are two types of priors used in this work. This figure shows a visualization of the tuned prior over weight $p(w|\phi)$. To show this prior, we plot the approximate posterior function $p(\theta | x, w)$ with $w \sim p(w)$. This is then before SBI training, as during SBI training, we compute the posterior over weights $p(w | D, \phi)$. Approximate posterior samples after SBI training would then be $p(\theta | x, w), w \sim p(w | D, \phi)$. We added some precisions in the caption to avoid any confusion.
>
> > In general, it seems that a better explanation of the steps involved in setting up and training the BNN-NPE (NRE) approach. I suggest adding more details in the next and adding an algorithm scheme in the text or appendix.
>
> We added a few sentences at the beginning of Section 3 indicating the different steps involved. There are three main steps. The first step is to optimize an appropriate prior $p(w | \phi)$. The second step is to use that prior to compute a posterior over weights $p(w | \phi, D)$ and the third step is to use that posterior over weight to compute an approximate posterior over simulator's parameters $\hat{p}(\theta | x) = \int p(\theta | x, w) p(w | D, \phi) dw$.

---

> > ### Comment · Reviewer_qwfa · 2024-11-18
> > **Response to rebuttal**
> >
> > Thank you for addressing my concern about the missing related work and benchmark citations. I appreciate the inclusion of Lueckmann et al. (2017) in the discussion.
> > And thanks for clarifying the use of different priors in the BNN approach. The visualization in Figure 1 is clear now as well.

---

> ### Author Response · Authors · 2024-11-14
> **Rebuttal**
>
> > the experimental results are difficult to interpret because they show the media of only three repetitions. More repetitions and error bars would be better here.
>
> We agree that more runs would have been preferable. However, we estimated the computational cost of 3 runs to be approximately 25,000 GPU hours. This is due to the fact that each point in the reported curves corresponds to 3 full training and testing procedures and that testing with Bayesian neural networks involves the approximation of the Bayesian model average for each test sample. Unfortunately, we do not have the computational power to do many more runs. That being said, although a bit noisy, we belive that those 3 runs are sufficient to demonstrate that using BNNs with our prior leads to better coverages in the low-simulation regime, which is our main claim.
>
> > the high underconfidence of the BNN approach in how-data regimes is concerning. Additional evaluation of the posterior predictive distributions would be appropriate.
>
> We would argue that it depends on what we are trying to achieve. This work follows the philosophy highlighted by Hermans et al., 2022, where they argue that to be useful for scientific application, posterior approximations must be conservative before anything else. If that is not the case, then they cannot be used reliably for downstream scientific tasks. In this work, the main objective is to build SBI methods that are conservative for low simulation budgets. Of course, we still wish to maintain reasonable statistical power but only if conservativeness is observed in the first place. In that regard, overconfidence is a much bigger issue than underconfidence and we mainly aim to avoid overconfidence. We agree that in some settings, people might not care that much about conservativeness and focus on statistical power. Our work does not aim to address this setting and other methods might be preferred for such applications.
>
> Joeri Hermans, Arnaud Delaunoy, Francois Rozet, Antoine Wehenkel, Volodimir Begy, and Gilles Louppe. A crisis in simulation-based inference? beware, your posterior approximations can be unfaithful. Transactions on Machine Learning Research, 2022
>
> > the choice and construction of the prior from simulated should be explained more clearly. A better explanation ideally will resolve the questions on the general procedure and on Figure 1 above.
>
> We hope our clarifications above made everything clearer. We dedicated an entire page (section 3.2) to this. Could you point out specific elements that would be interesting to add or clarify in your opinion?
>
> > NPE / NRE ensembles seems to perform similarly well compared to the BNN approach. How does it perform on the cosmology example? How does it compare to the BNN approach in terms of computational budget? Do you think ensembles could be a good alternative when computational budgets are limited? What are the disadvantages of ensembles compared to BNN for SBI?
>
> > More generally, what is the computational complexity of the BNN approach compared to NPE (ensembles)?
>
> From Figure 2, we can conclude that they do not perform similarly well, as the NPE ensemble has bad coverage for low simulation budgets, which is the setting of interest in this work, on all the benchmarks. BNN-NPE does not show the same behavior. We believe that the same observations would be made on the cosmological examples as it has been observed on all the benchmarks. In terms of computational budget, ensembles need more computational power to train than Bayesian neural networks with variational inference. For an ensemble of $N$ members, $N$ training procedure have to be performed, while for BNNs, only one training procedure outputing the posterior on weights has to be performed. At inference time, approximating the Bayes model average requires $M$ neural network evaluation if $M$ neural networks are used for the Monte-Carlo approximation. The two methods are then equivalent on that point if $M=N$.
>
> ---
>
> Overall, we think we have addressed each concern raised in your review and implemented changes accordingly. While these improvements appear to resolve the identified issues, we'd value your perspective on whether any aspects still fall short of expectations. If the changes adequately address your concerns, we'd appreciate having this reflected in the evaluation.

---

> > ### Comment · Reviewer_qwfa · 2024-11-18
> > **Response to rebuttal**
> >
> > Thank you for addressing my concern regarding the number of repetitions in the experimental results. I understand the significant computational cost involved in running more repetitions. However, given that all reviewers have pointed out the limited evaluation and have concerns regarding the performance of the BNN approach on some of the benchmarks, it would be important to address this. Running the evaluation five times and reporting SEM error bars would greatly enhance the transparency of your findings. I appreciate your efforts and understand the constraints, but additional runs would provide stronger evidence for your claims.

---

> > > ### Author Response · Authors · 2024-11-18
> > >
> > > Thank you for acknowledging the improvements regarding the citations, the clarifications of the priors used, and the clarification of Figure 1. While we answer the only remaining concern below, we would like to invite you to already reflect on those changes in your evaluation, which you can do by editing the review.
> > >
> > > We understand that 3 runs might sound insufficient intuitively. However, if we have a look at Figures 11 and 12, we can conclude that over 24 runs (3 runs on 4 benchmarks for 2 methods using BNNs based on the same prior), the use of BNNs with our prior leads to positive or only slightly negative coverage AUC. With that fact in mind, it becomes clear to us that this behavior over 24 runs in different settings cannot be pure luck and that our method is indeed working. If, on top of that, we wanted very smooth plots, then we believe that 5 runs would not be sufficient and that orders of magnitude more runs would be needed. This is, unfortunately, not possible. We hope you understand our point of view on that and that you are now convinced from the empirical evidence that our method is indeed working even if the plots are not the smoothest.

---

> ### Comment · Reviewer_qwfa · 2024-11-19
>
> Thank you for the additional clarifications. I will take your responses into account for the later discussion among the reviewers and AC.
>
> In the meantime, if you really cannot run additional experiments, could you please show the results with SEM error bars (instead of median)? This will provide more transparency regarding the empirical results - thanks!

---

> ### Author Response · Authors · 2024-11-20
>
> Here are the plots with the SEM errors plotted along the mean https://imgur.com/a/AZ1aots . We also invite you to see Figures 11 and 12, which contain similar information as they show the 3 runs and, hence, the minimum, median, and maximum values obtained.
>
> Thanks for engaging in the discussion and taking our responses into account. If that improved your opinion about the paper, could you reflect this in your evaluation?

---

> > ### Comment · Reviewer_qwfa · 2024-11-22
> >
> > Thank you for providing the additional plots.
> >
> > Overall, I agree with reviewer 9hZV on the contribution and limitations of the presented approach. It is a new promising approach to obtain conservative posteriors in low data regimes, but the results are difficult to interpret for now. This evaluation is reflected in my current score.
> >
> > Thank you for the detailed rebuttal. I am confident that we will converge to a fair decision during the upcoming discussion among the reviewers.

---

### Meta-Review · Area_Chair_YuPZ · 2024-12-19

**Metareview:**

The authors propose a low-budget Bayesian Neural Network (BNN) designed to provide accurate posterior estimates when limited data is available due to the high cost of simulations. The reviewers commended the algorithm and noted that the paper was generally easy to follow. However, they found the results insufficiently robust to support the claims made in the paper. Both the reviewers and the Area Chair acknowledged the challenge of obtaining more samples, which is a key motivation for this work. Nevertheless, the bar for acceptance as a new, reliable method is higher than simply demonstrating utility for a specific problem. While the results with 10 samples show slight improvements, the performance with only three samples could be a fluke. Moreover, as the number of samples increases, the proposed method fails to match the quality of state-of-the-art results.

**Additional Comments On Reviewer Discussion:**

THe authors and reviewers engage during the rebuttal period. However, the reviewers were unconvinced by the authors arguments. I side with the reviewers.

---

### Decision · Program_Chairs · 2025-01-22

Reject